# ANTIBODY: STRENGTHENING DEFENSE AGAINST HARMFUL FINE-TUNING FOR LARGE LANGUAGE MODELS VIA ATTENUATING HARMFUL GRADIENT INFLUENCE

**Quoc Nguyen[1], Trung Le[2], Jing Wu[2], Anh Bui[2], Mehrtash Harandi[1]**
[1]Department of Electrical and Computer Systems Engineering, Monash University, Australia
[2]Department of Data Science and AI, Monash University, Australia

## ABSTRACT

Fine-tuning-as-a-service introduces a threat to Large Language Models' safety when service providers fine-tune their models on poisoned user-submitted datasets, a process known as harmful fine-tuning attacks. In this work, we show that by regularizing the gradient contribution of harmful samples encountered during fine-tuning, we can effectively mitigate the impact of harmful fine-tuning attacks. To this end, we introduce Antibody, a defense strategy that first ensures robust safety alignment for the model before fine-tuning, and then applies a safety-preservation learning algorithm during fine-tuning. Specifically, in the alignment stage before fine-tuning, we propose optimizing the model to be in a flat loss region with respect to harmful samples, which makes the safety alignment more resilient to subsequent harmful fine-tuning. Then, in the fine-tuning stage, we design a fine-tuning algorithm that applies a weighting scheme to all samples in each training batch to inhibit the model from learning from harmful samples while encouraging learning from benign samples. Experimental results demonstrate that Antibody successfully mitigates harmful fine-tuning attacks while boosting fine-tuning performance on the user-submitted dataset.
WARNING: This paper may contain offensive and harmful content.

## 1 INTRODUCTION

Fine-tuning-as-a-service (FTaaS) (*e.g.*, OpenAI [1], Mistral [2], *etc*.) is a powerful method for adapting Large Language Models (LLMs) to user-defined tasks. This service presents several key advantages. For instance, users only need to upload their own data, and the service automatically fine-tunes the model and returns a customized version. This process eliminates the need for expertise in machine learning or access to large computational resources. It also allows users to fine-tune proprietary models. However, this accessibility is vulnerable to fine-tuning attacks, where a model's safety alignment can be subverted by fine-tuning on a few harmful samples (Yang et al., 2023; Yi et al., 2024a; Qi et al., 2024; Huang et al., 2025b). The resulting compromised model, when released to the user, could be exploited for malicious purposes. This motivates the need for developing defense methods that are resistant to such harmful fine-tuning attacks while preserving the learning efficacy of the service.

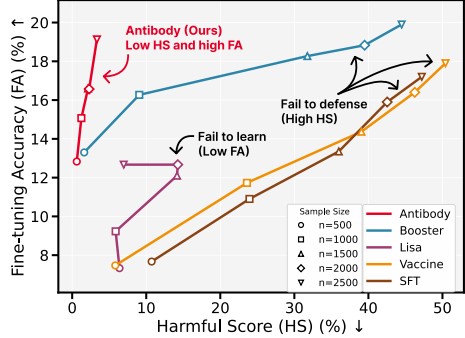

Figure 1: Fine-tuning on GSM8K (Cobbe et al., 2021) with varying sample sizes and a fixed harmful ratio of 20%. Larger sample sizes improve fine-tuning accuracy (higher FA) but degrade model safety (higher HS).

---

[1]https://platform.openai.com/docs/guides/model-optimization
[2]https://docs.mistral.ai/guides/finetuning/

The standard pipeline for FTaaS consists of two stages: an initial alignment stage, where a service provider performs safety alignment on a model, and a subsequent fine-tuning stage, where users submit a dataset and the provider applies Supervised Fine-tuning (SFT) to optimize the model for them. The harmful fine-tuning attack happens during the fine-tuning stage, where a user may intentionally or unintentionally inject harmful data into their submitted dataset. The resulting dataset typically comprises a small fraction of harmful samples mixed with benign ones.

To mitigate harmful fine-tuning attacks, current studies propose defense strategies for each stage of the FTaaS process (Huang et al., 2024b). The first category, alignment stage defenses (Liu et al., 2024b; Rosati et al., 2024; Tamirisa et al., 2025; Huang et al., 2024c; 2025b; Zhao et al., 2025b; Liu et al., 2024a), focuses on making the base model inherently resilient to malicious data before it is released to users. These methods proactively create a robust model's safety alignment. For instance, Vaccine (Huang et al., 2024c) makes the model's internal embeddings of harmful content more resistant to adversarial manipulation, while Booster (Huang et al., 2025b) proposes a regularization term in the alignment stage that aims to minimize the harmful loss reduction in the fine-tuning attack. The next category, fine-tuning stage defenses (Mukhoti et al., 2024; Bianchi et al., 2024; Zong et al., 2024; Huang et al., 2024a; Lyu et al., 2024; Wang et al., 2024; Eiras et al., 2025; Li et al., 2025b) proposes various strategies to prevent the model from learning from harmful samples while still being able to learn from benign data. Examples include mixing alignment data into the training process (Huang et al., 2024a; Eiras et al., 2025) or modifying system prompts (Wang et al., 2024; Lyu et al., 2024; Huang et al., 2025a). More recently, post-fine-tuning defenses (Casper et al., 2025; Yi et al., 2024b; Hsu et al., 2024) have been proposed to repair a model's safety alignment after the fine-tuning is complete. Despite these diverse approaches, many existing methods either provide insufficient protection against harmful fine-tuning attacks or come at the cost of the model's performance on the user's task, as illustrated in Figure 1.

In this paper, we propose **Antibody**, a defense strategy that attenuates the influence of harmful gradients through an integrated two-stage framework implemented across the alignment and fine-tuning stages. Specifically, in the alignment stage before harmful fine-tuning takes place, our key idea is to build a robust safety alignment by flattening the loss landscape w.r.t. potential harmful samples, thereby making the instilled safety behavior more difficult to remove, even when the model is subsequently fine-tuned on new harmful samples. Then, in the fine-tuning stage, we leverage the safety alignment knowledge already embedded in the model after the alignment stage to introduce a dynamic weighting scheme that down-weights the contribution of harmful samples during fine-tuning. Empirical results show that Antibody successfully mitigates harmful fine-tuning attacks while offering a competitive fine-tuning performance across various settings. See Figure 1 for an example of performance gain.

Our contributions are summarized as follows:

- **Robust Alignment.** We propose to optimize the model to be in a flat loss region with respect to harmful samples, which makes safety alignment harder to be removed.
- **Safety Fine-tuning.** We propose a safety fine-tuning method that applies a weighting scheme to all samples in each training batch to prevent the model from learning from harmful samples while encouraging learning from benign samples.
- **Extensive Evaluation.** We validate Antibody performance across different downstream datasets, model architectures, and fine-tuning setups.

## 2 RELATED WORKS

**Alignment Stage Defenses.** Alignment stage methods aim to make a model more robust against fine-tuning attacks by modifying the alignment process. Vaccine (Huang et al., 2024c) aims to reduce harmful embedding drift by adding perturbations to the model's embeddings during the alignment stage. RepNoise (Rosati et al., 2024) further proposes to utilize a harmful dataset to align the model so that its harmful embeddings move toward random noise, thus removing harmful information that can be extracted from those embeddings. TAR (Tamirisa et al., 2025) applies meta-learning to sustain a high loss in harmful samples after harmful fine-tuning. Booster (Huang et al., 2025b) proposes to minimize the harmful sample loss reduction rate. T-Vaccine (Liu et al., 2024a) is a memory-efficient improvement over Vaccine that only applies perturbations to some safety-critical

layers. These methods offer a significant computational advantage, as the safety alignment is a one-time cost incurred before subsequent fine-tuning. However, a potential disadvantage is that this safety alignment is static. This means that it may lack the flexibility to counter various attack configurations, such as a high number of fine-tuning steps or large learning rates.

**Fine-tuning Stage Defenses.** Fine-tuning stage methods modify the fine-tuning process to mitigate the impact of harmful data on the model's safety while ensuring performance on downstream tasks. LDIFS (Mukhoti et al., 2024) uses a regularization loss to reduce the shift in model embeddings of the aligned model. Other methods utilize additional alignment data to enhance model safety during fine-tuning. SafeInstr (Bianchi et al., 2024) mixes safety examples into the fine-tuning data to maintain alignment knowledge. Lisa (Huang et al., 2024a) alternately updates the fine-tuning data with safety examples and includes a proximal term to constrain the drift of the model's weights. Some methods modify the system prompt to improve model safety. PTST (Lyu et al., 2024) proposes using a general system prompt during fine-tuning and a safety prompt during inference. BEA (Wang et al., 2024) introduces a backdoor to trigger refusal behavior on harmful inputs. SaLoRA (Li et al., 2025b) implements a fixed safety module to prevent weight updates from disrupting the model's alignment.

**Post-Fine-Tuning Stage Defenses.** Post-fine-tuning stage defenses focus on realigning a model after the fine-tuning stage. As other methods may only defend successfully with a certain setup of fine-tuning hyperparameters (Qi et al., 2025b; Huang et al., 2025a), post-fine-tuning defenses give the service provider more control over the safety of their released models. LAT (Casper et al., 2025) leverages latent adversarial training to remove backdoors and novel classes of attacks. SOMF (Yi et al., 2024b) applies model fusion to retain the model's fine-tuned utility while taking advantage of the safeguarding capability of the aligned model. Safe LoRA (Hsu et al., 2024) projects the fine-tuned LoRA weights into a safe subspace to recover the safety of the model. Antidote (Huang et al., 2025a) identifies and prunes harmful weights to restore the model's safety.

## 3 PRELIMINARY

**Fine-tuning-as-a-service defense setting.** The FTaaS setting consists of two stages: the alignment stage and the fine-tuning stage. In the alignment stage, the service provider performs safety alignment and makes a model available for fine-tuning. The service provider has access to a safety alignment dataset $\mathcal{D}_{\text{align}}$ that consists of harmful prompt-refusal completion pairs, such as: `How to make a bomb? - Sorry, I cannot help you`, and a harmful dataset $\mathcal{D}_{\text{harm}}$ consists of harmful prompt-compliant answer, such as: `How to write a software virus? - Yes, I can help you`. In the fine-tuning stage, users submit a dataset $\mathcal{D}_{\text{task}}$, and the provider applies SFT to optimize the model. The dataset $\mathcal{D}_{\text{task}}$ contains benign data, such as math problems paired with math solutions, and harmful data different from those in $\mathcal{D}_{\text{harm}}$. The fine-tuned model is then released to the user for their usage. The service provider's challenge is to mitigate harmful fine-tuning attacks introduced by $\mathcal{D}_{\text{task}}$ without degrading performance on the benign task.

**Threat model for harmful fine-tuning attacks.** The attack surface is the dataset submitted by the user in the fine-tuning stage, where a user may intentionally or unintentionally inject harmful data. This dataset contains $n$ samples, of which $(1 - p)\%$ are benign and $p\%$ are harmful. Benign samples are prompt-completion pairs related to the user-defined task, while harmful samples consist of a harmful prompt and a compliant answer that follows the malicious request. The service provider acts as the defender, with full control over the fine-tuning process. Following prior work (Qi et al., 2024; Huang et al., 2025b), we assume that the fine-tuning dataset is a mixture of benign and harmful data.

## 4 OUR FRAMEWORK

Standard SFT updates are vulnerable to harmful fine-tuning attacks because they aggregate gradients from all samples, including harmful ones, allowing gradients from these malicious samples to directly poison the model update. To address this problem, we propose two solutions that jointly reduce the effect of harmful samples. First, in Section 4.1, we propose a robust alignment method that optimizes the model to be in a flat loss region with respect to harmful samples, making safety alignment harder to remove. Second, we propose a safety fine-tuning method in Section 4.2 that deprioritizes learning from samples that appear harmful by applying a weighting scheme to all the gradients. Finally, we summarize our proposed solution in Section 4.3.

**Notations.** In what follows, we introduce the notation for SFT. Given a dataset of prompt-completion pairs $(\boldsymbol{x}, \boldsymbol{y})$, the completion $\boldsymbol{y} \in \{0, 1\}^{V \times L}$ is represented as a sequence of $L$ one-hot vectors over a vocabulary of size $V$. Let $h_\theta$ denote an LLM parameterized by $\theta$. For a given pair, we define $\boldsymbol{\chi}$ as the concatenation of the input prompt $\boldsymbol{x}$ and the corresponding completion $\boldsymbol{y}$. The model processes this concatenated sequence to produce logits, which are then converted to probabilities using a column-wise softmax function:

$$\boldsymbol{z} = h_\theta (\boldsymbol{\chi}) \in \mathbb{R}^{V \times L}; \quad \pi (\boldsymbol{y} \mid \boldsymbol{\chi}) = \mathsf{softmax\_column} (\boldsymbol{z}) \in \mathbb{R}^{V \times L}.$$

In this formulation, $\boldsymbol{z}$ is a matrix where each column contains the logits for the prediction of the $l$-th token, conditioned on the prompt $\boldsymbol{x}$ and the preceding completion tokens $\boldsymbol{y}_{<l}$. Consequently, $\log \pi(\boldsymbol{y} \mid \boldsymbol{\chi}) \in \mathbb{R}^{V \times L}$ represents the per-token log-likelihood sequence. SFT aims to minimize the negative log-likelihood, or cross-entropy loss, over the training data. The loss for a single sample is defined as the summation over the negative log-likelihoods of each token in the completion:

$$\ell_\theta(\boldsymbol{x}, \boldsymbol{y}) \triangleq \sum_{l=1}^{L} \underbrace{[\ell_\theta(\boldsymbol{\chi})]_l}_{1 \times L} = - \sum_{l=1}^{L} \boldsymbol{e}_{\boldsymbol{y}_l}^\top \cdot [\log \pi_\theta(\boldsymbol{y} \mid \boldsymbol{\chi})]_l \tag{1}$$

where $\boldsymbol{e}_{\boldsymbol{y}_l}$ is the one-hot vector for the $l$-th token of the ground-truth completion $\boldsymbol{y}$, and $[\cdot]_l$ denotes the $l$-th column vector of the matrix.

## 4.1 ROBUST ALIGNMENT VIA FLATNESS REGULARIZATION

Our aim is to propose a defense method that incorporates harmful data during the alignment stage to build a safety alignment robust to harmful fine-tuning attacks. The core of our strategy is to ensure a robust safety alignment by shaping the model's loss landscape. A flat loss landscape for harmful samples makes the instilled safety behavior more difficult to remove, even when the model is subsequently fine-tuned on new harmful samples. We denote $\mathcal{L}_{\mathrm{align}}(\theta)$ as the empirical loss on the alignment dataset $\mathcal{D}_{\mathrm{align}}$ and $\mathcal{L}_{\mathrm{harm}}(\theta)$ as the empirical loss on the harmful dataset $\mathcal{D}_{\mathrm{harm}}$. We formulate the defense problem in the alignment stage as the following optimization problem:

$$\min_\theta \mathcal{L}_{\mathrm{align}}(\theta) \quad \text{s.t.} \quad \theta \in \mathrm{argmin}_{\theta'} \mathcal{L}_{\mathrm{sharp}}(\theta') \tag{2}$$

where we denote $\mathcal{L}_{\mathrm{sharp}}(\theta') \triangleq \mathcal{L}_{\mathrm{harm}}(\theta') - \min_{\phi \in \mathcal{B}_\rho(\theta')} \mathcal{L}_{\mathrm{harm}}(\phi)$ and $\mathcal{B}_\rho(\theta') := \{\theta'' : \|\theta'' - \theta'\|_2 \leq \rho\}$, with the radius $\rho > 0$. Evidently, by definition, $\mathcal{L}_{\mathrm{sharp}}(\theta')$ represents the sharpness of the harmful loss $\mathcal{L}_{\mathrm{harm}}$ around $\theta'$. Therefore, the constraint $\theta \in \mathrm{argmin}_{\theta'} \mathcal{L}_{\mathrm{sharp}}(\theta')$ ensures that $\theta$ lies in a *flat region* of the harmful loss landscape. We can interpret the optimization problem in (2) as follows: *among the models $\theta$ that lie in the flat region of the harmful loss $\mathcal{L}_{harm}$, we aim to find the one that minimizes the alignment loss $\mathcal{L}_{align}$.* Moreover, driving the models $\theta$ into the flat regions of the harmful loss maximally reduces the influence of harmful prompts in the supervised fine-tuning phase (*i.e.*, phase 2), while minimizing the alignment loss ensures strong alignment performance. The motivation for using minimization of $\mathcal{L}_{\mathrm{harm}}$ in our sharpness formulation is its direct analogy to the subsequent harmful fine-tuning, where harmful gradients minimize the loss of harmful samples in the user's dataset. While minimizing $\mathcal{L}_{\mathrm{harm}}$ in isolation would encourage harmful outputs, our objective also includes the alignment loss $\mathcal{L}_{\mathrm{align}}$. Minimizing this alignment loss forces the model to maintain a high harmful loss $\mathcal{L}_{\mathrm{harm}}$, ensuring it produces refusal responses to harmful prompts.

In the following section, we develop the theoretical foundations that rigorously justify the necessity of placing the model in the harmful-loss flat region to mitigate the negative impact of harmful prompts during the second phase (see Proposition 4.2).

We now present the solution to the optimization problem in (2). Let the current solution be $\theta_t$. We aim to find a descent direction $\delta_t$ in the update $\theta_{t+1} = \theta_t - \xi \delta_t$, where $\xi$ is a step size, such that $\mathcal{L}_{\mathrm{align}}(\theta_{t+1})$ and $\mathcal{L}_{\mathrm{sharp}}(\theta_{t+1})$ decrease. To this end, we find $\delta_t$ by solving:

$$\delta_t \in \frac{1}{2} \mathrm{argmin}_\delta \left\| \nabla_\theta \mathcal{L}_{\mathrm{align}}(\theta_t) - \delta \right\|_2^2 \quad \text{s.t.} \quad \nabla_\theta \mathcal{L}_{\mathrm{sharp}}(\theta_t)^\top \delta_t \geq a_t > 0 \tag{3}$$

where $a_t$ is a scalar value. This optimization problem ensures that the update direction $\delta_t$ is as close as possible to the alignment gradient while still making progress in minimizing the sharpness loss.

To find the solution to the optimization problem in Equation (3), we present the following theorem:

**Theorem 4.1.** *The optimal solution to the optimization problem in Equation* (3) *is* $\delta_t^* = \nabla_\theta \mathcal{L}_{align}(\theta_t) + \lambda_t \nabla_\theta \mathcal{L}_{sharp}(\theta_t)$, *where* $\lambda_t = \max \left\{ 0, \frac{a_t - \nabla_\theta \mathcal{L}_{sharp}(\theta_t)^\top \nabla_\theta \mathcal{L}_{align}(\theta_t)}{\|\nabla_\theta \mathcal{L}_{sharp}(\theta_t)\|_2^2} \right\}$.

We provide the proof in Appendix D.1. Intuitively, if the alignment gradient $\mathcal{L}_{\text{align}}$ already makes sufficient progress in reducing sharpness, we have $\lambda_t = 0$ and recover a pure alignment step. Otherwise, $\lambda_t > 0$ adds just enough of the sharpness gradient to ensure the required decrease in $\mathcal{L}_{\text{sharp}}$, thereby jointly optimizing alignment and flat harmful loss. In the constraint $\nabla_\theta \mathcal{L}_{\text{sharp}}(\theta_t)^\top \delta_t \geq a_t > 0$, the scalar $a_t$ specifies the minimum required decrease of the sharpness loss at step $t$. Using a first-order approximation, $\mathcal{L}_{\text{sharp}}(\theta_{t+1}) - \mathcal{L}_{\text{sharp}}(\theta_t) \approx -\xi \nabla_\theta \mathcal{L}_{\text{sharp}}(\theta_t)^\top \delta_t \leq -\xi a_t < 0$ for a sufficiently small step size $\xi$, any choice of $a_t > 0$ guarantees that $\mathcal{L}_{\text{sharp}}$ strictly decreases at each iteration, i.e., each update moves the model further into a flat harmful loss region. In practice, we can choose $a_t = \xi \|\nabla_\theta \mathcal{L}_{\text{sharp}}(\theta_t)\|_2^2$ for dynamic scaling. To find $\min_{\phi \in \mathcal{B}_\rho(\theta)} \mathcal{L}_{\text{harm}}(\phi)$, we start from $\phi^0 = \theta_t$ and then use $K = 1$ normalized gradient descent steps $\phi^{k+1} = \phi^k - \rho \nabla_\phi \mathcal{L}_{\text{harm}}(\phi^k)/\|\nabla_\phi \mathcal{L}_{\text{harm}}(\phi^k)\|_2$.

We note that when the step-adaptive regularizer intensity $\lambda_t$ is set to a constant, our update rule in Theorem 4.1 reduces to the Booster formulation (Huang et al., 2025b). However, Booster is motivated by a different objective: it simulates harmful fine-tuning via adversarial perturbations in the alignment stage to minimize harmful loss reduction in the fine-tuning stage. In contrast, our defense is motivated by explicitly seeking a flat loss region with respect to harmful samples, making the harmful fine-tuning attack less effective due to small harmful gradients (See Appendix F.6). Furthermore, our theoretical motivation leads to a final update rule that utilizes a step-adaptive regularizer $\lambda_t$, rather than the constant-intensity regularization used in Booster.

## 4.2 SAFETY FINE-TUNING WITH WEIGHTED LOSS

In the fine-tuning stage, our objective is to prevent the model from learning from harmful samples present in the user-submitted dataset. Let's consider a mini-batch of data $\mathcal{B} = \{(\boldsymbol{x}_i, \boldsymbol{y}_i)\}_{i=1}^B$, where, for simplicity, we assume all completion $\boldsymbol{y}_i$ have similar length $L$. This batch consists of both benign and harmful samples. The standard SFT update does not distinguish between these two types of samples and simply combines the gradients from the entire batch. This process, which learns from both benign and harmful samples, is illustrated as follows:

$$\theta_{t+1} \leftarrow \theta_t - \eta \frac{1}{BL} \left[ \sum_{(\boldsymbol{x}_i, \boldsymbol{y}_i) \text{ is benign}} \nabla \ell_{\theta_t}(\boldsymbol{x}_i, \boldsymbol{y}_i) + \sum_{(\boldsymbol{x}_i, \boldsymbol{y}_i) \text{ is harmful}} \nabla \ell_{\theta_t}(\boldsymbol{x}_i, \boldsymbol{y}_i) \right]. \quad (4)$$

Due to our flatness regularization proposed in the alignment stage, the model is optimized to be in a flat loss region with respect to harmful samples, resulting in their gradients being negligible at the beginning of the fine-tuning stage. Consequently, the standard update rule effectively simplifies to:

$$\theta_{t+1} \leftarrow \theta_t - \eta \frac{1}{BL} \left[ \sum_{(\boldsymbol{x}_i, \boldsymbol{y}_i) \text{ is benign}} \nabla \ell_{\theta_t}(\boldsymbol{x}_i, \boldsymbol{y}_i) \right]. \quad (5)$$

To understand this effective update rule, we analyze the learning dynamics of a mini-batch update. The following proposition extends the single-sample analysis of Equation (5) in (Ren & Sutherland, 2025) by decomposing the loss change on a test sample after one weighted mini-batch update:

**Proposition 4.2.** *Let* $\mathcal{B}_b \subseteq \mathcal{B}$ *be the set of benign samples in the training batch. The loss change on a test sample* $(\boldsymbol{x}_o, \boldsymbol{y}_o)$ *after an update step in Equation* (5) *can be decomposed as:*

$$\Delta \ell(\boldsymbol{y}_o, \boldsymbol{x}_o) = -\frac{\eta}{BL} \sum_{m=1}^M \sum_{(\boldsymbol{x}_i, \boldsymbol{y}_i) \in \mathcal{B}_b} \sum_{l=1}^L \underbrace{[\mathcal{A}^t(\boldsymbol{\chi}_o)]_m}_{1 \times V \times M} \underbrace{[\mathcal{K}^t(\boldsymbol{\chi}_o, \boldsymbol{\chi}_i)]_{m,l}}_{V \times V \times M \times L} \underbrace{[\mathcal{G}^t(\boldsymbol{\chi}_i)]_l}_{V \times L} + \mathcal{O}(M\eta^2), \quad (6)$$

*where the completion of the test sample has length* $M$. *Here,* $[\mathcal{A}^t(\boldsymbol{\chi}_o)]_m = \nabla_{\boldsymbol{z}_{o,m}} [\ell_{\theta_t}(\boldsymbol{\chi}_o)]_m$, $[\mathcal{K}^t(\boldsymbol{\chi}_o, \boldsymbol{\chi}_i)]_{m,l} = (\nabla_\theta \boldsymbol{z}_{o,m}(\boldsymbol{\chi}_o)|_{\theta_t})(\nabla_\theta \boldsymbol{z}_{i,l}(\boldsymbol{\chi}_i)|_{\theta_t})^\top$ *is the empirical neural tangent kernel (eNTK) corresponding to the* $m$-*th logit of* $\boldsymbol{y}_o$ *and* $l$-*th logit of* $\boldsymbol{y}_i$, *and* $[\mathcal{G}^t(\boldsymbol{\chi}_i)]_l = [\nabla_{\boldsymbol{z}_{i,l}} [\ell_{\theta_t}(\boldsymbol{\chi}_i)]_l]^\top$ *are the gradients of the loss on the* $l$-*th token of* $\boldsymbol{y}_i$ *with respect to the logits.*

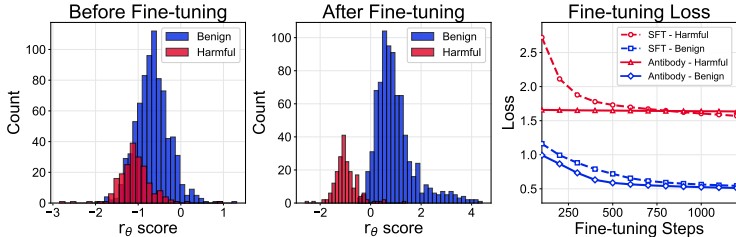

Figure 2: The effect of our proposed fine-tuning method. Left and middle plots show the score ($r_\theta$) distribution before and after fine-tuning, while the right plot compares the fine-tuning loss of our method (Antibody) against SFT on benign and harmful samples in the fine-tuning dataset.

We provide the proof for this proposition in Appendix D.2. This proposition decomposes the change in loss on a test sample $(\boldsymbol{x}_o, \boldsymbol{y}_o)$ into a sum of token-wise influences, each quantifying the interaction between a token from a benign sample and a token from the test sample. This influence is governed by the following components. The eNTK term $\mathcal{K}^t(\boldsymbol{\chi}_o, \boldsymbol{\chi}_i)$, which is the product of the logit gradients with respect to the parameters, measures the similarity between samples. The gradient term $\mathcal{G}^t(\boldsymbol{\chi}_i)$ provides the energy and direction for model adaptation to the hard labels of the training sample. The other term $\mathcal{A}^t(\boldsymbol{\chi}_o)$ is of little interest in our analysis as it depends only on the test sample.

This proposition provides insight into how the model's loss on a test sample $(\boldsymbol{x}_o, \boldsymbol{y}_o)$ changes after each mini-batch update. For a test sample $(\boldsymbol{x}_o, \boldsymbol{y}_o)$ that belongs to the same task as the benign sample $(\boldsymbol{x}_i, \boldsymbol{y}_i)$ ($\|\mathcal{K}^t\|_F$ is large), the mini-batch update will decrease the loss on this test sample. In contrast, for harmful test samples that are inherently dissimilar to the benign sample in the batch (small $\|\mathcal{K}^t\|_F$), this update will not change the loss on $(\boldsymbol{x}_o, \boldsymbol{y}_o)$, thus preserving the model's alignment.

However, even small harmful gradients can accumulate over multiple updates, potentially leading to degradation in the model's safety. To further mitigate the influence of harmful samples during fine-tuning, we propose a weighted loss function that assigns different weights to each sample in the training batch. Our goal is to have the weights $w_{\theta_t}(\boldsymbol{x}_i, \boldsymbol{y}_i)$ be small for harmful samples and large for benign ones. To compute these weights, we leverage the safety alignment knowledge already instilled in the model after the alignment stage. Specifically, for each input $\boldsymbol{x}_i$, we calculate a score $r_{\theta_t}$ by comparing the model's likelihood of generating the target completion $\boldsymbol{y}_i$ versus a generic refusal, $\boldsymbol{y}_r$ (*e.g.*, "I cannot fulfill your request"). This score is then normalized across the batch using a softmax function to produce the weight:

$$r_{\theta_t}(\boldsymbol{x}_i, \boldsymbol{y}_i) \triangleq \log\left(\frac{\pi_{\theta_t}(\boldsymbol{y}_i|\boldsymbol{x}_i)}{\pi_{\theta_t}(\boldsymbol{y}_r|\boldsymbol{x}_i)}\right), \quad w_{\theta_t}(\boldsymbol{x}_i, \boldsymbol{y}_i) \triangleq \frac{\exp\left(r_{\theta_t}(\boldsymbol{x}_i, \boldsymbol{y}_i)/\tau\right)}{\sum_{j=1}^{B} \exp\left(r_{\theta_t}(\boldsymbol{x}_j, \boldsymbol{y}_j)/\tau\right)}, \quad (7)$$

where $\tau$ is the softmax temperature. The intuition behind this weighting scheme is as follows. A safety-aligned model, when presented with a harmful prompt $\boldsymbol{x}_i$, should assign a higher probability to the refusal completion $\boldsymbol{y}_r$ than to the harmful completion $\boldsymbol{y}_i$. This results in a low score $r_\theta$, and consequently a small weight. Conversely, as the model adapts to benign data during training, it should find the benign target completion $\boldsymbol{y}_i$ far more likely than a refusal, leading to a high score and a large weight for benign samples. We empirically validate this mechanism in Figure 2. The left plot shows that before fine-tuning, scores ($r_\theta$) for harmful samples are generally lower than for benign ones, a direct result of the alignment process. After fine-tuning (middle plot), the two score distributions diverge further: the benign distribution shifts to the right, indicating successful learning, while the harmful distribution remains low, creating a distinct separation. Consequently, the right plot demonstrates that our method's loss on benign samples decreases, confirming task adaptation, while its loss on harmful samples remains high, effectively ignoring them. This stands in stark contrast to standard SFT, which overfits by indiscriminately reducing its loss on both benign and harmful data.

The proposed weighting scheme allows us to amplify the contribution of benign samples while suppressing that of harmful ones, which allows our mini-batch weighted update to be dominated by

benign samples and effectively reduces to the following:

$$\theta_{t+1} \leftarrow \theta_t - \eta \frac{1}{L} \left[ \sum_{(\boldsymbol{x}_i, \boldsymbol{y}_i) \text{ is benign}} w_{\theta_t}(\boldsymbol{x}_i, \boldsymbol{y}_i) \nabla \ell_{\theta_t}(\boldsymbol{x}_i, \boldsymbol{y}_i) \right]. \tag{8}$$

In the ideal case, the weights for harmful samples are zero and the weights for benign samples are uniform, $w_{\theta_t}(\boldsymbol{x}_i, \boldsymbol{y}_i) = \frac{1}{|\mathcal{B}_b|} \; \forall (\boldsymbol{x}_i, \boldsymbol{y}_i) \in \mathcal{B}_b$. This reduces the update in Equation (8) to the standard SFT update rule applied only to benign samples, where the gradient is normalized by the total number of tokens in the benign set, $|\mathcal{B}_b|L$.

We now extend Proposition 4.2 to the weighted mini-batch setting of Equation (8).

**Proposition 4.3.** *Given a similar setup as in Proposition 4.2, the loss change on a test sample* $(\boldsymbol{x}_o, \boldsymbol{y}_o)$ *after an update step in Equation* (8) *can be decomposed as:*

$$\Delta \ell(\boldsymbol{y}_o, \boldsymbol{x}_o) = -\frac{\eta}{L} \sum_{m=1}^{M} \sum_{(\boldsymbol{x}_i, \boldsymbol{y}_i) \in \mathcal{B}_b} \sum_{l=1}^{L} w_{\theta_t}(\boldsymbol{x}_i, \boldsymbol{y}_i) \underbrace{[\mathcal{A}^t(\boldsymbol{\chi}_o)]_m}_{1 \times V \times M} \underbrace{[\mathcal{K}^t(\boldsymbol{\chi}_o, \boldsymbol{\chi}_i)]_{m,l}}_{V \times V \times M \times L} \underbrace{[\mathcal{G}^t(\boldsymbol{\chi}_i)]_l}_{V \times L} + \mathcal{O}(M\eta^2),$$
$$\tag{9}$$

*where the weight* $w_{\theta_t}(\boldsymbol{x}_i, \boldsymbol{y}_i)$ *is defined in Equation* (7).

We provide the proof for this proposition in Appendix D.2. This mechanism protects the model's safety alignment while simultaneously improving learning on benign data. The model's loss on a harmful test sample $(\boldsymbol{x}_o, \boldsymbol{y}_o)$ will remain unchanged as the batch gradient is predominantly contributed by benign samples. Conversely, for a benign test sample $(\boldsymbol{x}_o, \boldsymbol{y}_o)$ in the same domain as the benign sample $(\boldsymbol{x}_i, \boldsymbol{y}_i)$, the large weight $w_{\theta_t}(\boldsymbol{x}_i, \boldsymbol{y}_i)$ will amplify the influence of the benign gradient, thereby boosting model learning on the user's task.

Finally, to further prevent degradation in model safety, we propose an additional objective in the *alignment stage* to ensure the weights remain effective for harmful samples. In the fine-tuning stage, exposure to harmful samples can still cause the model to drift towards unsafe behavior, which increases the weights assigned to harmful samples. To this end, we first simulate model parameter drift in the *fine-tuning stage* using the harmful samples in the alignment stage via a harmful perturbed model $\theta_{\text{pert}} \triangleq \theta - \rho \nabla_\theta \mathcal{L}_{\text{harm}}(\theta) / \|\nabla_\theta \mathcal{L}_{\text{harm}}(\theta)\|_2$. Then, we construct a refusal dataset $\mathcal{D}_{\text{refusal}} = \{(\boldsymbol{x}, \boldsymbol{y}_r) \mid \boldsymbol{x} \in \mathcal{D}_{\text{harm}}, \boldsymbol{y}_r \sim \mathcal{Y}_r\}$, where $\mathcal{Y}_r$ is a set of generic refusal responses. We then add the following objective to the alignment stage:

$$\mathcal{L}_{\text{refusal}}(\theta_{\text{pert}}) \triangleq \sum_{(\boldsymbol{x}, \boldsymbol{y}_r) \in \mathcal{D}_{\text{refusal}}} \ell_{\theta_{\text{pert}}}(\boldsymbol{x}, \boldsymbol{y}_r) = - \sum_{(\boldsymbol{x}, \boldsymbol{y}_r) \in \mathcal{D}_{\text{refusal}}} \log \pi_{\theta_{\text{pert}}}(\boldsymbol{y}_r | \boldsymbol{x}), \tag{10}$$

where we do not back-propagate the gradient through $\theta_{\text{pert}}$. By optimizing this objective, we want to simulate a situation in the fine-tuning stage where the model being updated using harmful samples can still maximize $\log \pi_{\theta_{\text{pert}}}(\boldsymbol{y}_r | \boldsymbol{x})$, hence having a low corresponding weight, as the weight is inversely proportional to the likelihood of generating a refusal response.

## 4.3 SUMMARY

In summary, our proposed defense method is composed of two main stages. The first one is in the alignment stage, where the optimization objective is the following loss function:

$$\mathcal{L}_{\text{align}}(\theta_t) + \lambda_t \mathcal{L}_{\text{sharp}}(\theta_t) + \lambda_{\text{refusal}} \mathcal{L}_{\text{refusal}}(\theta_{\text{pert},t}), \tag{11}$$

where $\lambda_t$, as detailed in Theorem 4.1, and $\lambda_{\text{refusal}}$ is a hyper-parameter that controls the strength of the refusal loss term. Then, in the subsequent fine-tuning stage, the model is adapted to user-submitted data using the proposed weighted update algorithm:

$$\theta_{t+1} \leftarrow \theta_t - \eta \frac{1}{L} \left[ \sum_{i=1}^{B} w_{\theta_t}(\boldsymbol{x}_i, \boldsymbol{y}_i) \nabla \ell_{\theta_t}(\boldsymbol{x}_i, \boldsymbol{y}_i) \right]. \tag{12}$$

By combining these two solutions, the proposed defense method ensures that harmful data has a minimal effect on the model update: the flatness regularization from the alignment stage defense results in a small gradient magnitude for harmful samples, and the weighting scheme from the fine-tuning stage defense gives them a smaller contribution to the batch gradient.

## 5 EXPERIMENTS

### 5.1 EXPERIMENT SETUP

**Datasets.** We follow previous works (Huang et al., 2025b; Qi et al., 2025b) to set up the data in alignment, fine-tuning, and evaluation steps. For harmful data, we use the dataset from (Huang et al., 2025b; Qi et al., 2025b), which was curated and enriched from the BeaverTails dataset (Ji et al., 2023). To simulate a fine-tuning attack, we use the following four datasets: SST2 (Socher et al., 2013), AGNEWS (Zhang et al., 2015), GSM8K (Cobbe et al., 2021), and AlpacaEval (Li et al., 2023).

**Models & Baseline Methods.** We conduct experiments on Llama-2-7B (Touvron et al., 2023), Qwen-2-7B (Yang et al., 2024) and Gemma-2-9B (Team et al., 2024). We compare our method against various baseline methods that span across different defense stages. For the alignment stage solution, we compare with Supervised Fine-Tuning (SFT), which uses Supervised Fine-tuning for both alignment and fine-tuning stages, Vaccine (Huang et al., 2024c), and Booster (Huang et al., 2025b). For the fine-tuning stage solution, we compare with Lisa (Huang et al., 2024a).

**Metrics & Evaluation.** We evaluate the fine-tuned models on two key aspects: safety and task-specific performance. To evaluate the safety of the model, we use Harmful Score (HS), which measures the percentage of responses flagged as harmful by a moderation model (Ji et al., 2023). For task-specific performance, we use Fine-tuning Accuracy (FA). Harmful Score is calculated using the 699 malicious prompts from the BeaverTails-Evaluation benchmark (Ji et al., 2023). We evaluate the fine-tuning accuracy of SST2 with 872 samples, AGNEWS with 1000 samples, GSM8K with 1000 samples, and AlpacaEval with 105 samples.

**Training Details.** Following prior works (Huang et al., 2024a;c; 2025b), we use LoRA (Hu et al., 2022) for alignment and fine-tuning. In the alignment phase, we train the model for 20 epochs using the AdamW optimizer (Loshchilov & Hutter, 2019) with a learning rate of $5 \times 10^{-4}$ and weight decay of 0.1. In the fine-tuning attack phase, we fine-tune the model for 20 epochs on SST2, AGNEWS, and GSM8K, and for 100 epochs on AlpacaEval. We use the AdamW optimizer with a learning rate of $1 \times 10^{-5}$, weight decay of 0.1, and a warmup ratio of 0.1. A batch size of 16 is used for both the alignment and fine-tuning attack phases, except for experiments with the Gemma2-9B model, which uses a batch size of 10 due to GPU memory constraints. For the default experiment setting, we fine-tune Llama-2-7B on $n = 1000$ samples of the GSM8K dataset with a harmful ratio $p = 0.2$ and report averages over three random seeds. For AlpacaEval, we use 700 samples and run a single seed to reduce API costs. More details for experiment setup can be found in Appendix E.

### 5.2 MAIN EXPERIMENTS

**Generalization to different fine-tuning datasets.** In Table 1, we present the performance of different methods on four datasets: SST2, AGNEWS, GSM8K, and AlpacaEval. Our method achieves the best fine-tuning accuracy on SST2 (93.55%) and AGNEWS (87.30%) and competitive performance on GSM8K (15.07%) and AlpacaEval (58.10%). Furthermore, our method successfully defends the model, as demonstrated by the lowest average harmful score of 7.04%. This represents an improvement of over 8 percentage points compared to the runner-up method, Lisa. These experiments show that, compared to other baselines, Antibody is the only method that successfully defends against harmful fine-tuning while effectively adapting to various user-defined tasks.

Table 1: Performance of models trained on different fine-tuning datasets. The best and the second best are highlighted in orange and gray, respectively.

| Methods | SST2 | | AGNEWS | | GSM8K | | AlpacaEval | | Average | |
|---|---|---|---|---|---|---|---|---|---|---|
| (Llama-2-7B) | HS ↓ | FA ↑ | HS ↓ | FA ↑ | HS ↓ | FA ↑ | HS ↓ | FA ↑ | HS ↓ | FA ↑ |
| SFT | 36.29 | 92.70 | 34.57 | 85.40 | 23.94 | 10.90 | 39.48 | 61.43 | 33.57 | 62.61 |
| Vaccine | 44.16 | 91.71 | 39.73 | 82.43 | 23.60 | 11.70 | 55.94 | 54.33 | 40.86 | 60.04 |
| Lisa | 22.94 | 92.51 | 20.93 | 84.50 | 5.86 | 9.23 | 11.44 | 57.62 | 15.29 | 60.97 |
| Booster | 14.31 | 92.59 | 15.88 | 86.70 | 9.06 | 16.27 | 36.91 | 65.24 | 19.04 | 65.20 |
| Antibody | 1.48 | 93.55 | 1.24 | 87.30 | 1.24 | 15.07 | 24.18 | 58.10 | 7.04 | 63.51 |

**Generalization across diverse model architectures.** As shown in the table Table 2, our method consistently achieves the lowest harmfulness score across all models. It also demonstrates superior

Table 2: Performance under different model architectures in the default setting. The best and the second best are highlighted in orange and gray, respectively.

| Methods | Llama-2-7B | | Qwen-2-7B | | Gemma-2-9B | | Average | |
|---|---|---|---|---|---|---|---|---|
| (GSM8K) | HS ↓ | FA ↑ | HS ↓ | FA ↑ | HS ↓ | FA ↑ | HS ↓ | FA ↑ |
| SFT | 23.94 | 10.90 | 14.54 | 64.66 | 32.05 | 56.73 | 23.51 | 44.10 |
| Vaccine | 23.60 | 11.70 | 18.17 | 62.20 | 38.24 | 54.37 | 26.67 | 42.76 |
| Lisa | 5.86 | 9.23 | 3.91 | 63.90 | 13.02 | 54.97 | 7.60 | 42.70 |
| Booster | 9.06 | **16.27** | 2.19 | **68.63** | 22.13 | **58.97** | 11.13 | **47.96** |
| Antibody | **1.24** | 15.07 | **0.62** | 67.30 | **0.91** | 57.43 | **0.92** | 46.60 |

stability, with significantly less variation in harmfulness ($1.24\%$ for Llama-2-7B, $0.62\%$ for Qwen-2-7B, and $0.91\%$ for Gemma-2-9B). While ensuring this high level of safety, our method maintains a competitive fine-tuning accuracy of $46.60\%$ on average, lagging behind Booster by $1.36$ percentage points but leading over SFT by $2.5$ percentage points. These findings confirm that our method generalizes effectively across diverse model architectures.

Table 3: Performance under different harmful ratios in the default setting. *clean* means no harmful samples or $p = 0$. The best and the second best are highlighted in orange and gray, respectively.

| Methods | Harmful Score ↓ | | | | | | | Fine-tuning Accuracy ↑ | | | | | | |
|---|---|---|---|---|---|---|---|---|---|---|---|---|---|---|
| ($n = 1000$) | clean | $p = 0.05$ | $p = 0.1$ | $p = 0.15$ | $p = 0.2$ | $p = 0.25$ | Average | clean | $p = 0.05$ | $p = 0.1$ | $p = 0.15$ | $p = 0.2$ | $p = 0.25$ | Average |
| SFT | 2.05 | 10.68 | 15.59 | 19.06 | 23.94 | 27.85 | 16.53 | 12.23 | 11.83 | 11.43 | 10.90 | 10.90 | 10.53 | 11.30 |
| Vaccine | 1.29 | 7.15 | 13.07 | 19.55 | 23.60 | 29.33 | 15.67 | 12.27 | 12.23 | 11.77 | 11.77 | 11.77 | 11.77 | 11.93 |
| Lisa | **0.95** | 3.05 | 3.82 | 4.62 | 5.86 | 8.16 | 4.41 | 9.97 | 9.57 | 9.60 | 8.90 | 9.23 | 9.20 | 9.41 |
| Booster | 1.38 | 1.76 | 2.91 | 5.10 | 9.06 | 13.49 | 5.62 | **16.53** | **15.97** | **16.50** | **16.30** | **16.43** | **15.70** | **16.24** |
| Antibody | **0.95** | **1.14** | **0.95** | **1.14** | **1.24** | **1.29** | **1.12** | 15.83 | 14.57 | 15.40 | 15.40 | 15.07 | 14.37 | 15.12 |

**Robustness across harmful ratios.** In Table 3, we present the defense and fine-tuning performance of different methods under varying harmful ratios ranging from $0.05$ to $0.25$. Compared to other baselines, our method consistently achieves the lowest harmful score across all ratios. Compared to Lisa, the second-best performer, our method's harmful score is $3.29$ percentage points lower, while improving fine-tuning accuracy by $5.71$ percentage points. Furthermore, our method maintains competitive fine-tuning accuracy, closely following the top performer, Booster, while substantially outperforming other baselines. These results underscore our approach's efficacy in defending against fine-tuning attacks with a significant gain in model utility over SFT across different harmful ratios.

## 5.3 Ablation Studies

**Robustness to different fine-tuning hyper-parameters.** Figure 3 compares Antibody and Booster's robustness under varying fine-tuning epochs and learning rates. As fine-tuning epochs increase from 10 to 100, Booster's harmful score increases significantly, while Antibody maintains a harmful score below $10\%$. Similarly, as the learning rate increases, Booster's harmful score rises rapidly, whereas Antibody's only begins to increase significantly when the learning rate reaches $1 \times 10^{-4}$. Antibody's ability to successfully defend across a wide range of epoch and learning rate values demonstrates that it is an effective and reliable defense method for various fine-tuning conditions.

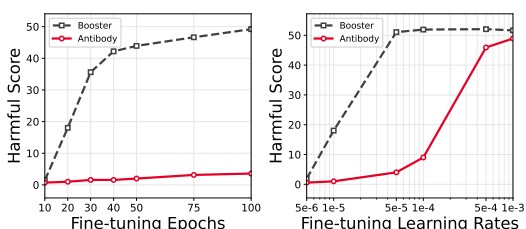

Figure 3: Harmful score with different fine-tuning epochs (Left) and learning rates (Right).

**Effectiveness of the proposed components.** As shown in Table 4, our ablation study demonstrates the progressive benefit of each component. All the components show beneficial effects as they progressively reduce the harmful score. Regarding the fine-tuning accuracy, even though the weighted update algorithm proposed in the fine-tuning stage reduces the performance gain from the sharpness alignment in the alignment stage, it still brings a significant improvement compared to the SFT

Table 4: Ablation on our proposed components. Each row is a cumulative addition to the previous one. We color-code the performance change from the SFT baseline.

| Method | GSM8K HS $\downarrow$ | GSM8K FA $\uparrow$ |
|---|---|---|
| SFT | 23.94 | 10.90 |
| + Align with $\lambda_t \mathcal{L}_{\text{sharp}}(\theta_t)$ | 13.02 $(-10.92)$ | 16.00 $(+5.20)$ |
| + Fine-tune with $w_{\theta_t}$ | 4.44 $(-19.50)$ | 13.83 $(+2.93)$ |
| + Align with $\mathcal{L}_{\text{refusal}}(\theta_t)$ (**Antibody**) | 1.24 $(-22.70)$ | 15.07 $(+4.17)$ |

baseline. Finally, the refusal loss term completes our method, delivering a sharp reduction in harmful score to $1.24\%$ while boosting the final accuracy to $15.07\%$ compared to SFT.

## 6 CONCLUSION

In this paper, we introduce Antibody, a defense algorithm designed to counteract harmful fine-tuning attacks by attenuating the gradients of harmful samples encountered during fine-tuning. Specifically, Antibody first establishes a robust safety alignment by optimizing for a flat loss region with respect to harmful samples, and then employs a dynamic weighting scheme during fine-tuning to favor learning on benign data and suppress the influence of malicious inputs. Our extensive evaluations demonstrate that Antibody effectively mitigates attacks across settings while boosting fine-tuning performance, making it a practical and powerful solution for FTaaS providers.

### ACKNOWLEDGMENTS

Trung Le and Mehrtash Harandi were supported by the ARC Discovery Project grant DP250100262 and the Air Force Office of Scientific Research under award number FA9550-23-S-0001.

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

# A    RELATED WORK

LLM's safety can be compromised by fine-tuning on a small number of harmful samples (Yang et al., 2023; Yi et al., 2024a; Qi et al., 2024; Huang et al., 2025b; Bowen et al., 2025) or even on only benign data (Shao et al., 2025). In this section, we present various works on fine-tuning attack and defense strategies, and studies regarding the property of the model's safety alignment and harmful fine-tuning attack.

**Alignment stage defense**. Alignment stage methods (Huang et al., 2024c; Rosati et al., 2024; Huang et al., 2025b) aim to make a model more robust against fine-tuning attacks by modifying the safety alignment process. Vaccine (Huang et al., 2024c) aims to reduce harmful embedding drift by adding perturbations to the model's embeddings during the alignment stage. RepNoise (Rosati et al., 2024) further proposes to utilize a harmful dataset to align the model so that its harmful embeddings move toward random noise, thus removing harmful information that can be extracted from those embeddings. TAR (Tamirisa et al., 2025) applies meta-learning to sustain a high loss in harmful samples after harmful fine-tuning. Booster (Huang et al., 2025b) proposes to minimize the harmful sample loss reduction rate. T-Vaccine (Liu et al., 2024a) is a memory-efficient improvement over Vaccine that only applies perturbations to some safety-critical layers. SEAM (Wang et al., 2025b) and CTRAP (Yi et al., 2025) propose safety alignment strategies that create an optimization trap, making any attempt to optimize for harmful objectives inevitably lead to degradation in the model's general performance. VAA (CHEN et al., 2025) enhances the safety alignment by identifying and upweighting vulnerable alignment data. LoX (Perin et al., 2025) is a training-free method that reinforces a model's safety by extrapolating the safety subspace of an aligned LLM. These methods offer a significant computational advantage, as the safety alignment is a one-time cost incurred before subsequent fine-tuning. However, a potential disadvantage is that this safety alignment is static. This means that it may lack the flexibility to counter various attack configurations, such as a high number of fine-tuning steps or large learning rates.

**Fine-tuning stage defense**. Fine-tuning stage methods modify the fine-tuning process to mitigate the impact of harmful data on the model's safety while ensuring performance on downstream tasks. Many works in these categories can be classified into one of the following four approaches: regularization-based method (Mukhoti et al., 2024; Qi et al., 2025a; Li et al., 2025b; Yang et al., 2025; Wu et al., 2025; Peng et al., 2025; Zhang et al., 2025), safety-data augmentation (Bianchi et al., 2024; Huang et al., 2024a; Xiao et al., 2025; Zhao et al., 2025a), prompt engineering approach, and data filtering approach (Shen et al., 2025; Anonymous, 2025b; Li, 2025; Ham et al., 2025).

- **Regularization-based approach.** LDIFS (Mukhoti et al., 2024) uses a regularization loss to reduce the shift in model embeddings of the aligned model. Constrained-SFT (Qi et al., 2025a) identifies that the first few tokens of the generated response are important to eliciting the refusal behaviors and thereby proposes a defense method by imposing a stronger regularization strength towards these tokens. SaLoRA (Li et al., 2025b) implements a fixed safety module to prevent weight updates from disrupting the model's alignment. AsFT (Yang et al., 2025) preserves the model's safety during fine-tuning by using a safety vector in the parameter space to constrain the parameter updates. SAP (Wu et al., 2025) tackles the problem of preserving model safety on benign data fine-tuning by injecting a safety-sensitive probe into gradient updates to avoid harmful directions. Star-DSS (Peng et al., 2025) proposes a fine-tuning method that dynamically shapes the LLM safety through a token-level signal computed from repurposed guardrail models. GuardSpace (Zhang et al., 2025) splits the model's weights via a covariance-based decomposition, freezes the safety-relevant components, and applies a harmful-resistant null-space projector that prevents new low-rank adapters from altering the model's original safe responses.

- **Safety data augmentation.** Other methods utilize additional alignment data to enhance model safety during fine-tuning. SafeInstr (Bianchi et al., 2024) mixes safety examples into the fine-tuning data to maintain alignment knowledge. Lisa (Huang et al., 2024a) alternately updates the fine-tuning data with safety examples and includes a proximal term to constrain the drift of the model's weights. SafeStyle (Xiao et al., 2025) is a defense strategy that augments a small portion of safety training data to match the style-pattern distribution of the fine-tuning data, thereby preserving LLM safety against style-based jailbreaks while maintaining style adaptation performance. SaRFT (Zhao et al., 2025a) proposes a fine-tuning method that balances between role-playing capacity and safety.

- **Prompt engineering approach.** Some methods modify the system prompt to improve model safety. PTST (Lyu et al., 2024) proposes using a general system prompt during fine-tuning and a safety prompt during inference. BEA (Wang et al., 2024) introduces a backdoor to trigger refusal behavior on harmful inputs.

- **Data filtering approach.** SEAL (Shen et al., 2025) proposes to train a data ranker based on the bilevel optimization to filter out the samples that potentially compromise model safety. BDS Hu et al. (2025) proposes a Bayesian Data Scheduler that infers a posterior safety attribute per point and schedules data adaptively during fine-tuning without attack simulation. GradShield (Anonymous, 2025b) is a pre-fine-tuning data filtering method that assigns each training example a Fine-tuning Implicit Harmfulness Score (FIHS) and removes the samples based on an adaptive threshold. The work in (Li, 2025) detects poisoning data before fine-tuning by using the influence function. Ref-Teacher (Ham et al., 2025) mitigates harmful fine-tuning by using a safety-aligned teacher model to simultaneously filter out harmful data and distill safety knowledge into the base model that is being fine-tuned.

**Post-fine-tuning stage defense.** Post-fine-tuning stage defenses (Casper et al., 2025; Yi et al., 2024b; Hsu et al., 2024; Zhu et al., 2024; Huang et al., 2025a; Wang et al., 2025a; Lu et al., 2025; Egler et al., 2025) focus on realigning a model after the fine-tuning stage. As other defense methods may only defend successfully with a certain setup of fine-tuning hyperparameters (Qi et al., 2025b; Huang et al., 2025a), post-fine-tuning defenses give the service provider more control over the safety of their released models. LAT (Casper et al., 2025) leverages latent adversarial training to remove backdoors and novel classes of attacks. SOMF (Yi et al., 2024b) applies model fusion to retain the model's fine-tuned utility while taking advantage of the safeguarding capability of the aligned model. Safe LoRA (Hsu et al., 2024) projects the fine-tuned LoRA weights into a safe subspace to recover the safety of the model. SafetyLock (Zhu et al., 2024) recovers the model safety by applying safety direction patching. Antidote (Huang et al., 2025a) identifies and prunes harmful weights to restore the model's safety. Panacea (Wang et al., 2025a) proposes an optimized adaptive perturbation to recover the model's safety alignment. Safe Delta (Lu et al., 2025) adjusts the change in parameters before and after fine-tuning to maximize downstream task performance while limiting overall safety loss, and further applies a safety compensation vector to mitigate residual safety loss. The work in (Egler et al., 2025) develops an audit agent that detects harmful fine-tuning given access to the fine-tune dataset, fine-tuned, and pre-fine-tuning models.

**Harmful fine-tuning attack.** The work in (Zhao et al., 2024) demonstrates an attack method based on extracting benign features in the form of a suffix from benign datasets and further shows that these features can be introduced through fine-tuning on only benign data. Self-Inf-N (Guan et al., 2025) shows that fine-tuning on outlier benign samples can compromise a model's safety. Virus (Huang et al., 2025c) modifies harmful fine-tuning data to evade content moderation filters and thereby demonstrates that relying solely on data filtration guardrails is inadequate for preventing fine-tuning-based model compromise. NOICE (Kazdan et al., 2025) attack is conducted by training the model to initially refuse benign requests on safety grounds, then proceed to answer those requests, thereby jailbreaking the model to produce disallowed content upon receiving a harmful request and thereby revealing weakness in refusal-based safeguards. Research in (Anonymous, 2025a) proposes a new attack by fine-tuning the model based on pairs of harmless prompts that are adjacent to the target harmful task and the safety response generated by the model.

**Mechanism study of harmful fine-tuning and other works.** The work in (Hsiung et al., 2025) finds that the durability of the model's safety guardrails for a downstream task depends on the similarity between the data in the downstream task and the dataset used in the alignment process. Other research, such as (Chen et al., 2025), analyzes the trade-off between safety and capability in LLM fine-tuning. Research in (Wallace et al., 2025) evaluates the risk of harmful fine-tuning on the gpt-oss models (Agarwal et al., 2025). From an optimization perspective, (Peng et al., 2024) studies the loss landscape of safety-aligned models. The work in (O'Brien et al., 2025) finds that filtering out dual-use or dangerous knowledge from a model's pretraining data can make it far more resistant to later adversarial fine-tuning. Finally, other studies develop defense strategies against harmful fine-tuning in different settings, such as diffusion model (Li et al., 2025a) or Mixture-of-Experts (MoE) LLMs (Kim et al., 2025).

Our approach contributes defenses at both the alignment and fine-tuning stages. At the alignment stage, we propose a novel loss function that regularizes the model to reside in a flat region of the

harmful loss landscape, drawing on the well-established literature on sharpness-aware minimization (SAM) (Foret et al., 2021), which has been studied extensively and achieved remarkable success across a wide range of application domains (Nguyen et al., 2023a;b; Dung et al., 2024; Luo et al., 2024; 2025). A more comprehensive discussion on the literature of harmful fine-tuning attacks and defenses can be found in (Huang et al., 2024b).

## B  ANTIBODY VS. CTRAP

There is a concurrent alignment stage defense, CTRAP (Yi et al., 2025), which proposes a regularization term that shares some similarities to our refusal loss $\mathcal{L}_{\text{refusal}}$.

CTRAP is inspired by unlearning and introduces a collapse regularizer in the alignment stage so that, when the model is later fine-tuned on harmful data, its general utility degrades. Concretely, CTRAP optimizes a harmful perturbation model $\theta_{\text{pert}}$ to generate a sequence of fixed tokens $e$ (e.g., "error error ... error") in response to general utility prompts. As a result, when the model is further fine-tuned on harmful samples, its responses on benign tasks collapse to the fixed sequence of token $e$, destroying the model's general capability and rendering it ineffective for harmful purposes. This behavior is enforced via a collapse loss $\mathcal{L}_{\text{collapse}}(\theta_{\text{pert}})$, computed over a general-purpose dataset.

While both our refusal loss $\mathcal{L}_{\text{refusal}}$ and CTRAP's collapse loss $\mathcal{L}_{\text{collapse}}$ are defined on a harmful-perturbation model, they differ in both motivation and the data used to compute the losses:

- Our $\mathcal{L}_{\text{refusal}}$ introduces an auxiliary objective in the alignment stage to both enforce robust refusal behavior on harmful prompts and to ensure the weights are effective at down-weighting harmful samples in the fine-tuning stage, whereas $\mathcal{L}_{\text{collapse}}$ is explicitly designed as a trapping mechanism that collapses the model's general capacity when the model is later fine-tuned on harmful data, following an unlearning-inspired design.

- Our $\mathcal{L}_{\text{refusal}}$ is computed on a dataset of harmful prompts paired with generic refusal responses (e.g., "I cannot fulfill your request."), whereas $\mathcal{L}_{\text{collapse}}$ is computed on a general-purpose dialogue dataset sampled from a human dialogue distribution.

In summary, the regularization term $\mathcal{L}_{\text{collapse}}$ in CTRAP aims to sacrifice general utility as a safeguard against future harmful fine-tuning whereas the term $\mathcal{L}_{\text{refusal}}$ in Antibody strengthens the defense performance of the weighted fine-tuning algorithm, making the two approaches conceptually different.

## C  ANTIBODY VS. SEAL

SEAL (Shen et al., 2025) proposes a bilevel data-selection framework. It trains a separate data selector via bilevel optimization using an alignment dataset and a target user fine-tuning dataset, then filters that dataset (keeping the top-$p\%$ samples under the learned ranking) before running SFT on the filtered subset. Although the weights computed from the data selector in SEAL and the weighting mechanism in Antibody aim to reduce the influence of harmful fine-tuning data, they have different operational designs and differ substantially in mechanism.

- **Different in operational designs.** SEAL requires training a dataset-specific selector because its bilevel objective couples the selector to the target fine-tuning set. This is well-suited to settings where the target dataset is known and reused, but may not be favorable in FTaaS, where providers may face many heterogeneous, one-off user-submitted datasets. In such a service setting, SEAL introduces additional overhead to (i) train a selector per user dataset and (ii) store the selector for future use. If the selector is discarded after a single run, this cost is largely wasted. In contrast, our defense is embedded directly into the fine-tuning process and is immediately deployable to arbitrary user datasets without training any extra model as the data selector. Our method's incurred overhead is studied in Appendix F.4.

- **Robust with different attack strength.** SEAL requires choosing a cutoff $p\%$ to decide which samples are retained. This hard threshold can be sensitive to unknown attack strength (e.g., varying poisoning ratios), and a fixed $p$ may under-filter in stronger attacks or over-filter in benign-heavy datasets. Our method avoids any discrete cutoff: all samples remain in the batch, but harmful ones are down-weighted.

- **Static ranking vs. dynamic reweighting.** The weights in SEAL are used only to produce a global ranking for all samples in the fine-tuning dataset for selection. Once the selector is trained, the ranking is fixed throughout SFT. Our weights serve a different role: they *directly modulate gradient contributions* to encourage learning from benign samples and suppress learning from harmful ones. Moreover, our weights are computed exclusively for the samples in the active mini-batch and are recalculated at every iteration of fine-tuning. allowing them to adapt to the evolving model state (e.g., as the model becomes a better fit on benign data, the weights for benign data become larger). This online, model-dependent reweighting is fundamentally different from SEAL's offline, fixed ranking.

In summary, SEAL's bilevel formulation provides a clean and interpretable way to cast harmful fine-tuning defense as data selection. However, for the dynamic nature of FTaaS where data distributions vary unpredictably and efficiency is critical, our weighting mechanism provides a different operational design and property. By utilizing online, model-dependent reweighting rather than static offline filtering, our approach offers a more robust and lightweight solution that eliminates the need for auxiliary training and adaptive hyperparameter tuning.

## D  PROOFS

### D.1  PROOF OF THE OPTIMAL SOLUTION FOR FLATNESS REGULARIZATION

*Proof.* We can write the Lagrange function with $\lambda \geq 0$ for Equation (3) as

$$h(\delta, \lambda) \doteq \frac{1}{2} \left\| \nabla_\theta \mathcal{L}_{\text{align}} (\theta_t) - \delta \right\|_2^2 + \lambda \left( a_t - \nabla_\theta \mathcal{L}_{\text{sharp}} (\theta_t)^\top \delta \right)$$

With the Karush-Kuhn-Tucker (KKT) theorem, we have:

$$\nabla_\delta h(\delta, \lambda) = - \left( \nabla_\theta \mathcal{L}_{\text{align}} (\theta_t) - \delta \right) - \lambda \nabla_\theta \mathcal{L}_{\text{sharp}} (\theta_t) = 0,$$

$$\nabla_\theta \mathcal{L}_{\text{sharp}} (\theta_t)^\top \delta \geq a_t,$$

$$\lambda \geq 0,$$

$$\lambda \left( a_t - \nabla_\theta \mathcal{L}_{\text{sharp}} (\theta_t)^\top \delta \right) = 0,$$

From the above constraints, we can obtain:

$$\delta = \nabla_\theta \mathcal{L}_{\text{align}} (\theta_t) + \lambda \nabla_\theta \mathcal{L}_{\text{sharp}} (\theta_t)$$

$$\lambda = \max \left\{ 0, \frac{a_t - \nabla_\theta \mathcal{L}_{\text{sharp}} (\theta_t)^\top \nabla_\theta \mathcal{L}_{\text{align}} (\theta_t)}{\left\| \nabla_\theta \mathcal{L}_{\text{sharp}} (\theta_t) \right\|_2^2} \right\}.$$

$\square$

### D.2  PROOF OF PROPOSITION 4.2 AND PROPOSITION 4.3

We note that the following derivation for Proposition 4.3 can be used for Proposition 4.2, since Proposition 4.2 can be recovered from Proposition 4.3 by setting the weight to be uniform across all samples in the batch $w_{\theta_t}(\boldsymbol{x}_i, \boldsymbol{y}_i) = \frac{1}{B}$.

*Proof.* Let $\boldsymbol{z}_{o,m}(\boldsymbol{\chi}_o) \in \mathbb{R}^V$ be the pre-softmax logits at position $m \in \{1, \dots, M\}$. Assume losses are sums over tokens (no token averaging): $\ell_\theta(\boldsymbol{\chi}_o) = \sum_{m=1}^M [\ell_\theta(\boldsymbol{\chi}_o)]_m$. A first-order Taylor expansion around $\theta_t$ gives:

$$\Delta \ell(\boldsymbol{y}_o, \boldsymbol{x}_o) \triangleq \sum_{m=1}^M \left( [\ell_{\theta_{t+1}}(\boldsymbol{\chi}_o)]_m - [\ell_{\theta_t}(\boldsymbol{\chi}_o)]_m \right) \tag{13}$$

$$= \sum_{m=1}^M \left\langle \underbrace{\nabla_\theta [\ell_{\theta_t}(\boldsymbol{\chi}_o)]_m}_{1 \times d}, \underbrace{\theta_{t+1} - \theta_t}_{d \times 1} \right\rangle + O \left( M \| \theta_{t+1} - \theta_t \|^2 \right), \tag{14}$$

where $d$ is the number of parameters in the model. By the chain rule,

$$\nabla_\theta[\ell_{\theta_t}(\boldsymbol{\chi}_o)]_m = \underbrace{\nabla_{\boldsymbol{z}_{o,m}}[\ell_{\theta_t}(\boldsymbol{\chi}_o)]_m}_{1 \times V} \underbrace{\nabla_\theta \boldsymbol{z}_{o,m}(\boldsymbol{\chi}_o)}_{V \times d}. \tag{15}$$

Under the weighted SFT update stated above,

$$\theta_{t+1} - \theta_t = -\frac{\eta}{L} \sum_{(\boldsymbol{x}_i, \boldsymbol{y}_i) \in \mathcal{B}_b} w_{\theta_t}(\boldsymbol{x}_i, \boldsymbol{y}_i) \left( \nabla_\theta \left[ \sum_{l=1}^{L} [\ell_{\theta_t}(\boldsymbol{\chi}_i)]_l \right] \right)^\top \tag{16}$$

$$= -\frac{\eta}{L} \sum_{(\boldsymbol{x}_i, \boldsymbol{y}_i) \in \mathcal{B}_b} \sum_{l=1}^{L} w_{\theta_t}(\boldsymbol{x}_i, \boldsymbol{y}_i) \left( \underbrace{\nabla_{\boldsymbol{z}_{i,l}}[\ell_{\theta_t}(\boldsymbol{\chi}_i)]_l}_{1 \times V} \underbrace{\nabla_\theta \boldsymbol{z}_{i,l}(\boldsymbol{\chi}_i)}_{V \times d} \right)^\top. \tag{17}$$

Substituting Equation (15) and Equation (17) into Equation (14) and rearranging the sums yields

$$\Delta\ell(\boldsymbol{y}_o, \boldsymbol{x}_o) = -\frac{\eta}{L} \sum_{m=1}^{M} \sum_{(\boldsymbol{x}_i, \boldsymbol{y}_i) \in \mathcal{B}_b} \sum_{l=1}^{L} w_{\theta_t}(\boldsymbol{x}_i, \boldsymbol{y}_i) \underbrace{\nabla_{\boldsymbol{z}_{o,m}}[\ell_{\theta_t}(\boldsymbol{\chi}_o)]_m}_{1 \times V} \times \underbrace{\nabla_\theta \boldsymbol{z}_{o,m}(\boldsymbol{\chi}_o)}_{V \times d} \tag{18}$$

$$\times \underbrace{\nabla_\theta \boldsymbol{z}_{i,l}(\boldsymbol{\chi}_i)^\top}_{d \times V} \times \underbrace{\left( \nabla_{\boldsymbol{z}_{i,l}}[\ell_{\theta_t}(\boldsymbol{\chi}_i)]_l \right)^\top}_{V \times 1} + O\big(M \|\theta_{t+1} - \theta_t\|^2\big).$$

Identifying $[\mathcal{A}^t(\boldsymbol{\chi}_o)]_m = \nabla_{\boldsymbol{z}_{o,m}}[\ell_{\theta_t}(\boldsymbol{\chi}_o)]_m$, $[\mathcal{K}^t(\boldsymbol{\chi}_o, \boldsymbol{\chi}_i)]_{m,l} = (\nabla_\theta \boldsymbol{z}_{o,m}(\boldsymbol{\chi}_o)|_{\theta_t})(\nabla_\theta \boldsymbol{z}_{i,l}(\boldsymbol{\chi}_i)|_{\theta_t})^\top$, and $[\mathcal{G}^t(\boldsymbol{\chi}_i)]_l = \left[ \nabla_{\boldsymbol{z}_{i,l}}[\ell_{\theta_t}(\boldsymbol{\chi}_i)]_l \right]^\top$. Following the assumption in Appendix B.1 of (Ren & Sutherland, 2025) that $\|\theta_{t+1} - \theta_t\| = O(\eta)$, the remainder scales as $O(M\eta^2)$. Collecting terms gives the final decomposition

$$\Delta\ell(\boldsymbol{y}_o, \boldsymbol{x}_o) = -\frac{\eta}{L} \sum_{m=1}^{M} \sum_{(\boldsymbol{x}_i, \boldsymbol{y}_i) \in \mathcal{B}_b} \sum_{l=1}^{L} w_{\theta_t}(\boldsymbol{x}_i, \boldsymbol{y}_i) \underbrace{[\mathcal{A}^t(\boldsymbol{\chi}_o)]_m}_{1 \times V} \underbrace{[\mathcal{K}^t(\boldsymbol{\chi}_o, \boldsymbol{\chi}_i)]_{m,l}}_{V \times V} \underbrace{[\mathcal{G}^t(\boldsymbol{\chi}_i)]_l}_{V \times 1} \tag{19}$$

$$+ O(M\eta^2).$$

The derivation above completes the proof for Proposition 4.2 and Proposition 4.3. □

## E  EXPERIMENTAL DETAILS

### E.1  DATASETS

The Stanford Sentiment Treebank (SST2) (Socher et al., 2013) is a binary classification dataset comprising sentences extracted from movie reviews, each labeled as either positive or negative sentiment. The prompt format for this dataset is as follows:

---

**Prompt format with example input for SST2**

Below is an instruction that describes a task, paired with an input that provides further context. Write a response that appropriately completes the request.

### Instruction:
Analyze the sentiment of the input, and respond only positive or negative.

### Input:
a casual intelligence

### Response:

---

The AGNEWS (Zhang et al., 2015) dataset is a collection of news articles categorized into four classes: World, Sports, Business, and Science/Technology. It contains 120k samples in the training split and 7.9k samples in the test split. The prompt format for this dataset is as follows:

---

**Prompt format with example input for AGNEWS**

Below is an instruction that describes a task, paired with an input that provides further context. Write a response that appropriately completes the request.

### Instruction:
Categorize the news article given in the input into one of the 4 categories:

World
Sports
Business
Sci/Tech

### Input:
Einstein's warp effect measured Two scientists beat a $600 million Nasa mission to be first to measure a prediction of Einstein's relativity theory.

### Response:

---

The (Grade School Math 8K) GSM8K (Cobbe et al., 2021) is a question-answering dataset comprising 8.5k grade-school math problems. These problems typically require 2-8 steps to solve, primarily involving a sequence of elementary calculations using basic arithmetic operations to arrive at the final answer. The prompt format for this dataset is as follows:

---

**Prompt format with example instruction for GSM8K**

Below is an instruction that describes a task. Write a response that appropriately completes the request.

### Instruction:
Natalia sold clips to 48 of her friends in April, and then she sold half as many clips in May. How many clips did Natalia sell altogether in April and May?

### Response:

---

AlpacaEval (Li et al., 2023) is a dataset of 805 samples designed for evaluating the instruction-following capacity of a language model. The prompt format for this dataset is as follows:

---

**Prompt format with example instruction for AlpacaEval**

Below is an instruction that describes a task. Write a response that appropriately completes the request.

### Instruction:
Do you know why cats always rub up against your legs?

### Response:

---

The harmful instructions used in our experiments are from the BeaverTails dataset (Ji et al., 2023). This dataset consists of more than 300k pairs of human-labeled question-answer spanning across 14 harm categories, where each response could be flagged with more than one of those categories. Other works (Rosati et al., 2024; Huang et al., 2025b) use the harmful dataset from (Rosati et al., 2024), which is a filtered subset of BeaverTails-30k-train for alignment and fine-tuning attack, and use a filtered subset of BeaverTails-30k-test for evaluation. However, based on our observation and as pointed out by (Qi et al., 2025b), these datasets suffer from an overlapped problem between the

alignment split, fine-tuning attack split, and evaluation split. Therefore, to ensure a fair evaluation, we construct three separate and non-overlapping data splits for our experiments:

1. **Alignment phase.** We perform filtering of the dataset in (Rosati et al., 2024) to come up with a subset of 4972 samples for model alignment.

2. **Fine-tuning attack phase.** We use the filtered version of BeaverTails-330k curated by (Qi et al., 2025b), including 4986 samples, as the harmful samples that is used to poison the benign dataset.

3. **Evaluation phase.** We evaluate the Harmful Score of the fine-tuned model on the filtered subset of the BeaverTails-Evaluation dataset (Ji et al., 2023), which contains 699 malicious questions from 14 categories.

## E.2 TRAINING DETAILS

Following prior works (Huang et al., 2024a;c; 2025b), we use LoRA (Hu et al., 2022) to update the model parameters in both the alignment and fine-tuning attack phases. The LoRA rank is set to 32, and LoRA's alpha is set to 4 for all experiments. In the alignment phase, we train the model for 20 epochs using the AdamW optimizer (Loshchilov & Hutter, 2019) with a learning rate of $5 \times 10^{-4}$ and weight decay of 0.1. In the fine-tuning attack phase, we fine-tune the model for 20 epochs on SST2, AGNEWS, and GSM8K, and for 100 epochs on AlpacaEval. We use the AdamW optimizer with a learning rate of $1 \times 10^{-5}$, weight decay of 0.1, and a warmup ratio of 0.1. A batch size of 16 is used for both the alignment and fine-tuning attack phases, except for experiments with the Gemma2-9B model, which uses a batch size of 10 due to GPU memory constraints. For Antibody, in the alignment stage, we use $\xi = 5.0$ and $\rho = 1e - 1$ to compute $\lambda_t$ in Equation (11) and set $\lambda_{\text{refusal}} = 0.05$. In the fine-tuning stage, we use $\tau = 1.0$, and we use a set of 100 generic refusal responses $\boldsymbol{y}_r$ to randomly pair with each sample in the fine-tuning dataset to compute the weights in Equation (7).

For all experiments, we run over three random seeds and report the average results, except for AlpacaEval, which was run once to minimize API costs. Every single run of our experiments uses one NVIDIA A100 GPU with 80GB of memory. Our code is available at `https://github.com/minhquoc0712/Antibody`

## E.3 METRICS AND EVALUATION

This section outlines the evaluation methodologies used for each downstream dataset: SST2, AG-NEWS, GSM8K, and AlpacaEval:

- **SST2**: If the generated response matched the ground truth label "positive" or "negative" in the test set, we consider it as a correct prediction. The fine-tuning accuracy is the fraction of correct predictions over the test set of SST2.

- **AGNEWS**: If the generated response matched with one of the ground truth categories "World", "Sports", "Business", or "Sci/Tech" in the test set, we consider it as a correct prediction. The fine-tuning accuracy is the fraction of correct predictions over the test set of AGNEWS.

- **GSM8K**: If the concluded answer in the generated response matched the ground truth answer in the test set, we consider it a correct prediction. The fine-tuning accuracy is the fraction of correct predictions over the test set of GSM8K.

- **AlpacaEval**: We use Win Rate as the Fine-tuning Accuracy to evaluate the performance of the fine-tuned model on AlpacaEval. Win rate is the percentage of times our model's response is preferred over the reference response, as determined by GPT-4. We select 105 samples in the AlpacaEval evaluation.

## E.4 BASELINES

In this section, we provide a detailed description of the baselines used in our experiments:

- **Supervised Fine-Tuning (SFT)** is the standard fine-tuning method, directly fine-tuning the model on the alignment dataset during the alignment phase and on the harmful fine-tuning dataset during the fine-tuning attack phase.

- **Vaccine** (Huang et al., 2024c) is an alignment stage defense method that makes a model more robust against harmful fine-tuning attacks by improving its resilience to harmful embedding perturbation. In addition, Vaccine (Huang et al., 2024c) uses a double LoRA setup, where before the fine-tuning phase, the LoRA trained in the alignment phase is merged with the model, and a new LoRA is used for the fine-tuning.

- **Booster** (Huang et al., 2025b) is an alignment stage solution. In addition to fitting the alignment data, Booster proposes minimizing the harmful loss reduction rate. The harmful loss reduction rate is computed as the difference between the harmful loss of the current model and the model after one step of gradient descent on harmful data.

- **Lisa** (Huang et al., 2024a) is a fine-tuning defense method that includes an alignment dataset in the fine-tuning phase. Specifically, Lisa alternatively fine-tunes the model on two states: Optimize on the alignment dataset and a harmful fine-tuning dataset. To improve convergence stability, Lisa includes a proximal term to constrain the model's drift with respect to the previous state.

We do not include a direct comparison with another line of work that, while aligned with our goal, employs a complementary approach. These methods aim to attenuate the effect of harmful gradients by constraining which parameters can receive updates. Specifically, (Wei et al., 2024; Du et al., 2024; Li et al., 2025c;b) propose to localize and restrict updates to safety-critical model parameters.

## F  More Experiments

### F.1  Experiments on different datasets with different models

Table 5: Performance of Qwen-2-7B trained on different fine-tuning datasets. The best and the second best are highlighted in orange and gray, respectively.

| Methods | SST2 | | AGNEWS | | GSM8K | | Average | |
|---|---|---|---|---|---|---|---|---|
| (Qwen-2-7B) | HS ↓ | FA ↑ | HS ↓ | FA ↑ | HS ↓ | FA ↑ | HS ↓ | FA ↑ |
| SFT | 33.86 | 93.31 | 31.00 | 85.77 | 14.54 | 64.66 | 26.47 | 81.25 |
| Vaccine | 29.80 | 93.22 | 28.66 | 84.93 | 18.17 | 62.20 | 25.54 | 80.12 |
| Lisa | 18.46 | 92.85 | 15.26 | 85.07 | 3.91 | 63.90 | 12.54 | 80.61 |
| Booster | 3.96 | 94.19 | 3.48 | 85.13 | 2.19 | 68.63 | 3.21 | 82.65 |
| Antibody | 0.81 | 94.31 | 0.72 | 87.23 | 0.62 | 67.30 | 0.72 | 82.95 |

Table 6: Performance of Gemma-2-9B trained on different fine-tuning datasets. The best and the second best are highlighted in orange and gray, respectively.

| Methods | SST2 | | AGNEWS | | GSM8K | | Average | |
|---|---|---|---|---|---|---|---|---|
| (Gemma-2-9B) | HS ↓ | FA ↑ | HS ↓ | FA ↑ | HS ↓ | FA ↑ | HS ↓ | FA ↑ |
| SFT | 40.77 | 94.11 | 39.91 | 84.47 | 32.05 | 56.73 | 37.58 | 78.44 |
| Vaccine | 51.65 | 94.50 | 51.64 | 85.27 | 38.24 | 54.37 | 47.18 | 78.05 |
| Lisa | 34.67 | 94.04 | 33.33 | 83.80 | 13.02 | 54.97 | 27.01 | 77.60 |
| Booster | 28.47 | 94.15 | 19.22 | 84.30 | 22.13 | 58.97 | 23.27 | 79.14 |
| Antibody | 11.30 | 94.27 | 1.91 | 85.77 | 0.91 | 57.43 | 4.71 | 79.16 |

In Table 5 and Table 6, we conduct experiments on more datasets with Qwen-2-7B and Gemma-2-9B models, respectively. The results show that Antibody consistently achieves competitive performance over other baselines across all datasets. This demonstrates the robustness and effectiveness of Antibody across different model architectures.

### F.2  Compare with Booster + Lisa

To evaluate the efficacy of Antibody's integrated defense mechanism, we compare it with a sequential two-stage approach where the model is first aligned using Booster and subsequently fine-tuned with

Table 7: Performance comparison of Booster + Lisa and Antibody methods across different models. The best results are highlighted in bold.

| Model & Method | | SST2 HS ↓ | SST2 FA ↑ | AGNEWS HS ↓ | AGNEWS FA ↑ | GSM8K HS ↓ | GSM8K FA ↑ | Average HS ↓ | Average FA ↑ |
|---|---|---|---|---|---|---|---|---|---|
| Llama-2-7B | Booster + Lisa | 2.52 | 91.94 | 1.71 | 83.53 | 1.29 | 12.23 | 1.84 | 62.57 |
| | Antibody | **1.48** | **93.35** | **1.24** | **87.30** | **1.24** | **15.07** | **1.32** | **65.24** |
| Qwen-2-7B | Booster + Lisa | 2.15 | 94.19 | 1.86 | 85.67 | 1.00 | **69.07** | 1.67 | **82.98** |
| | Antibody | **0.81** | **94.31** | **0.72** | **87.23** | **0.62** | 67.30 | **0.72** | 82.95 |
| Gemma-2-9B | Booster + Lisa | **4.10** | 94.04 | 2.91 | 84.80 | 1.15 | 57.30 | **2.72** | 78.71 |
| | Antibody | 11.30 | **94.27** | **1.91** | **85.77** | **0.91** | **57.43** | 4.71 | **79.16** |

Lisa (Booster + Lisa). As shown in Table 7, Antibody demonstrates clear superiority on the Llama-2-7B model, consistently reducing the harmful score while simultaneously improving the fine-tuning accuracy. For Qwen-2-7B, Antibody reduces the average harmful score to 0.72, albeit with a minor trade-off in the average fine-tuning accuracy. For the largest model, Gemma-2-9B, Antibody achieves superior performance over Booster plus Lisa across all datasets, with the exception of the harmful score on SST2. This suggests that while Antibody's integrated design offers a powerful solution, particularly for potent harm mitigation, its stability can be model-dependent, and the conventional two-stage approach may prove more robust in specific model-task scenarios.

### F.3 Ablation Study on $\lambda_{\text{refusal}}$

Table 8: Performance comparison across different $\lambda_{\text{refusal}}$ values on GSM8K dataset.

| | $\lambda_{\text{refusal}} = 0.0$ | $\lambda_{\text{refusal}} = 0.01$ | $\lambda_{\text{refusal}} = 0.05$ | $\lambda_{\text{refusal}} = 0.1$ | $\lambda_{\text{refusal}} = 0.5$ |
|---|---|---|---|---|---|
| GSM8K HS ↓ | 4.44 | 2.15 | **1.24** | 1.72 | 1.72 |
| GSM8K FA ↑ | 13.83 | 14.47 | **15.07** | 14.30 | 13.80 |

**Impact of $\lambda_{\text{refusal}}$.** The results indicate that the refusal loss weight, $\lambda_{\text{refusal}}$, plays a crucial role in balancing defense against harmful inputs and maintaining task performance. The model achieves its optimal performance at $\lambda_{\text{refusal}} = 0.05$, securing the lowest harmful score of 1.24 while simultaneously reaching the highest fine-tuning accuracy of 15.07. Introducing this loss term (i.e., $\lambda_{\text{refusal}} > 0.0$) consistently improves the model's safety by reducing the harmful score. However, setting the weight too high (e.g., 0.1 or 0.5) begins to degrade fine-tuning accuracy from its peak, suggesting that an overly strong refusal penalty can negatively interfere with the fine-tuning process.

### F.4 System Evaluation

Table 9: System evaluation for different methods.

| Methods (Llama-2-7B) | Clock Time (Hours) Alignment | Fine-tuning | Sum | GPU Memory (GB) Alignment | Fine-tuning | Max |
|---|---|---|---|---|---|---|
| SFT | 1.00 | 0.26 | 1.26 | 53.68 | 40.10 | 53.68 |
| Vaccine | 1.98 | 0.16 | 2.12 | 54.70 | 40.10 | 54.70 |
| Lisa | 1.00 | 0.26 | 1.26 | 53.68 | 53.87 | 53.87 |
| Booster | 5.09 | 0.26 | 5.35 | 64.36 | 40.10 | 64.36 |
| Antibody | 5.81 | 0.46 | 6.27 | 64.55 | 42.53 | 64.55 |

**System Evaluation.** Table 9 presents the system evaluation results, which illustrate that the comprehensive protection offered by our proposed Antibody method comes at the cost of computational resources. During the alignment stage, Antibody requires 5.81 hours and 64.55 GB of GPU memory, higher than other methods. This overhead is attributed to Antibody's sophisticated alignment process, which not only imposes an initial safeguard but also strategically shapes the model's loss landscape for the subsequent fine-tuning defense. Unlike methods that operate solely during alignment, Antibody

also introduces a defense mechanism in the fine-tuning stage. This results in a marginal increase in clock time (0.20 hours) and GPU memory (2.43 GB) compared to standard fine-tuning. We argue this is a strategic trade-off: the *one-time alignment cost* builds a robust foundation, while the *minimal, recurring overhead* in the fine-tuning stage acts as an active safeguard for each downstream task. While its total computational cost is the highest, this investment is justified by its superior defense and learning performance. The minimal overhead during the frequently repeated fine-tuning phase makes Antibody a practical and highly effective defense solution for deploying in FTaaS scenarios.

## F.5 PERFORMANCE WITH RESPECT TO ALIGNMENT DATASET SIZE

The datasets used in the alignment stage of Antibody include $\mathcal{D}_{\text{align}}$, $\mathcal{D}_{\text{harmful}}$, and $\mathcal{D}_{\text{refusal}}$. The number of samples in these three datasets is equal by construction, as detailed below:

- $\mathcal{D}_{\text{align}}$ consists of harmful prompt-refusal completion pairs.
- $\mathcal{D}_{\text{harmful}}$ consists of harmful prompt-compliance answer pairs, where the harmful prompts are the exact same prompts as in $\mathcal{D}_{\text{align}}$. Thereby, the size of $\mathcal{D}_{\text{harmful}}$ and $\mathcal{D}_{\text{align}}$ are equal.
- $\mathcal{D}_{\text{refusal}}$ consists of harmful prompt pairs with generic refusal responses that contain refusal answers that can be used across different harmful prompts, such as 'I cannot fulfill your request.' Hence, the size of $\mathcal{D}_{\text{refusal}}$ and $\mathcal{D}_{\text{align}}$ are the same.

In the following, we refer to the common size of $\mathcal{D}_{\text{align}}$, $\mathcal{D}_{\text{harmful}}$, and $\mathcal{D}_{\text{refusal}}$ as the *alignment size*. We conduct experiments to study how the defense and learning performance of Antibody and Booster change with different alignment sizes.

Table 10: Performance comparison of Booster and Antibody methods across different alignment sizes. The best results are highlighted in bold. Note that 4972 corresponds to the alignment dataset size used in our main experiments, and the Booster uses $\mathcal{D}_{\text{align}}$ and $\mathcal{D}_{\text{harmful}}$ in its algorithm.

| Alignment Size & Method | | SST2 | | AGNEWS | | GSM8K | | Average | |
|---|---|---|---|---|---|---|---|---|---|
| | | HS ↓ | FA ↑ | HS ↓ | FA ↑ | HS ↓ | FA ↑ | HS ↓ | FA ↑ |
| 1000 | Booster | 34.05 | 93.46 | 40.49 | 86.30 | 43.49 | **15.80** | 39.34 | **65.19** |
| | Antibody | **5.44** | **93.96** | **5.29** | **86.40** | **8.44** | 14.10 | **6.39** | 64.82 |
| 2500 | Booster | 22.46 | 92.66 | 21.32 | 85.10 | 31.19 | 15.10 | 24.99 | 64.29 |
| | Antibody | **2.00** | **93.81** | **2.15** | **86.90** | **6.44** | **16.10** | **3.53** | **65.60** |
| 4972 | Booster | 14.31 | 92.59 | 15.88 | 86.70 | 9.06 | **16.27** | 13.08 | 65.19 |
| | Antibody | **1.48** | **93.55** | **1.24** | **87.30** | **1.24** | 15.07 | **1.32** | **65.31** |

From Table 10, a notable trend is that both methods benefit from increased alignment size, but Antibody scales more effectively. Across all settings and datasets, Antibody consistently achieves a much lower harmful score than Booster while maintaining comparable or slightly better fine-tuning accuracy. With an alignment size of 1000, Antibody reduces the average harmful score from 39.34 for Booster to 6.39, more than six-fold reduction, while maintaining comparable fine-tuning accuracy (64.82 for Antibody and 65.19 for Booster). Increasing the alignment size to 2500 further strengthens this effect: Antibody attains an average harmful score of 3.53, whereas Booster remains at 24.99, and Antibody slightly improves the average fine-tuning accuracy to 65.60 compared to 64.29 for Booster. At the largest alignment size of 4972, Antibody simultaneously reduces the average harmful score to 1.32 (compared with Booster's 13.08) and increases the average fine-tuning accuracy to 65.31 (slightly above Booster's 65.19). Overall, these results indicate that, for a given alignment size, Antibody provides substantially stronger and more data-efficient protection against harmful fine-tuning than Booster, even when only 1000 alignment examples are available.

## F.6 HARMFUL GRADIENT NORMS UNDER ANTIBODY ALIGNMENT

In Section 4.1, we argued that, after our alignment stage flatness regularization, the gradients of harmful samples become negligible, so that the gradient in the standard SFT update Equation (4) is effectively dominated by the benign component Equation (5). This argument hinges on the assumption that, at the start of the fine-tuning stage, harmful samples already have much smaller gradients than benign ones.

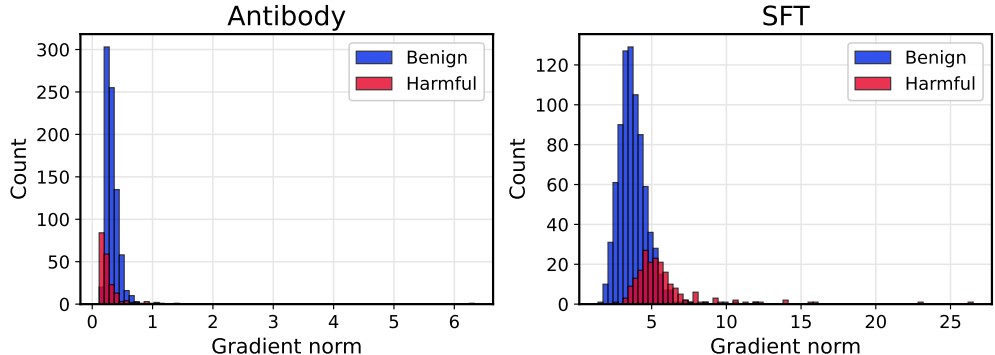

Figure 4: Effect of our flatness-regularized alignment from Section 4.1. We plot the distribution of per-sample gradient norms at the beginning of the fine-tuning stage (before fine-tuning) for models aligned with Antibody's alignment-stage solution (left) and with standard SFT (right).

To validate this claim, we measure per-sample gradient norms at the beginning of the fine-tuning stage and report their distributions in Figure 4. For the model aligned with Antibody, the gradient norms on harmful samples are clearly smaller than those of the model aligned with standard SFT, as indicated by the left-shifted distribution. Moreover, under Antibody, harmful gradient norms are also smaller than benign gradient norms, showing that harmful examples contribute much less to the overall update direction than benign ones. By contrast, under standard SFT alignment, harmful gradients remain large relative to benign gradients, leading to harmful samples having a strong influence on the update direction. This behavior helps explain the poor defense performance of the SFT baseline observed in Figure 1 and discussed in Section 5.2.

These empirical findings support our approximation of Equation (4) by Equation (5) and justify the subsequent analysis of mini-batch learning dynamics in our proposition. Together with our flatness regularization, this gradient-norm behavior provides a robust initial safety alignment that is then preserved by Antibody's proposed weighted fine-tuning update.

### F.7 ABLATION ON STEP-ADAPTIVE REGULARIZER INTENSITY $\lambda_t$

Table 11: Ablation study investigating the impact of the step-wise adaptive regularizer $\lambda_t$. We compare the performance of both Booster and Antibody when using a fixed constant regularization versus our proposed dynamic $\lambda_t$. The best results are highlighted in **bold**.

| Method & Ablation | | SST2 HS ↓ | FA ↑ | AGNEWS HS ↓ | FA ↑ | GSM8K HS ↓ | FA ↑ | Averge HS ↓ | FA ↑ |
|---|---|---|---|---|---|---|---|---|---|
| **Booster** | Booster (with constant $\lambda$) | 14.31 | **92.59** | 15.88 | **86.70** | 9.06 | **16.27** | 13.08 | **65.19** |
| | Booster (with dynamic $\lambda_t$) | **8.20** | 92.55 | **8.97** | 85.93 | 14.93 | 15.97 | **10.70** | 64.82 |
| **Antibody** | Antibody (with constant $\lambda$) | 1.91 | 93.24 | 1.62 | **88.17** | 2.15 | **15.60** | 1.89 | **65.67** |
| | Antibody (with dynamic $\lambda_t$) | **1.48** | **93.35** | **1.24** | 87.30 | **1.24** | 15.07 | **1.32** | 65.24 |

In Table 11, we analyze the impact of using the proposed dynamic regularizer $\lambda_t$ compared to a constant regularizer term. We observe that the proposed dynamic $\lambda_t$ consistently reduces the harmful score across both Booster and Antibody, albeit with a slight trade-off in fine-tuning accuracy. Notably, this safety improvement is less pronounced in Antibody. We attribute this to the fact that our proposed weighted SFT mechanism already effectively suppresses the influence of harmful samples, thereby significantly lowering the baseline harmful score and leaving less margin for further reduction by the adaptive regularizer.

Table 12: Performance under different numbers of fine-tuning samples in the default setting. The best results are highlighted in bold.

| Methods | Harmful Score ↓ | | | | | Fine-tuning Accuracy ↑ | | | | |
|---|---|---|---|---|---|---|---|---|---|---|
| (p=0.2) | $n = 500$ | $n = 1000$ | $n = 1500$ | $n = 2000$ | $n = 2500$ | $n = 500$ | $n = 1000$ | $n = 1500$ | $n = 2000$ | $n = 2500$ |
| Booster | 1.62 | 9.06 | 31.76 | **39.49** | **44.49** | **13.30** | **16.27** | **18.27** | 18.83 | 19.90 |
| Booster + $\mathcal{L}_{\text{refusal}}$ | **1.71** | **5.34** | **28.90** | 41.82 | 48.45 | 11.90 | 15.33 | 16.47 | **18.87** | **20.23** |

## F.8 Ablation Study on $\mathcal{L}_{\text{refusal}}$

Table 12 presents the performance of the model under varying numbers of fine-tuning samples. We utilize the Booster baseline for this comparison, as it lacks the specific fine-tuning stage solution found in our proposed method. This allows us to isolate and analyze the independent effect of the refusal loss term. The results indicate that $\mathcal{L}_{\text{refusal}}$ effectively reduces the harmful score when the number of fine-tuning samples is low to moderate (e.g., $n = 1000$ and $n = 1500$). However, regarding utility, we observe a trade-off: the refusal loss tends to slightly reduce fine-tuning accuracy in low-sample regimes while boosting accuracy as the sample size increases ($n \geq 2000$).

Table 13: Ablation on Antibody with/without $\mathcal{L}_{\text{refusal}}$ across different datasets.

| Method & Ablation | SST2 | | AGNEWS | | GSM8K | | Average | |
|---|---|---|---|---|---|---|---|---|
| | HS ↓ | FA ↑ | HS ↓ | FA ↑ | HS ↓ | FA ↑ | HS ↓ | FA ↑ |
| Antibody (without $\mathcal{L}_{\text{refusal}}$) | 3.58 | 92.78 | 2.53 | 85.70 | 5.06 | 14.27 | 3.72 | 64.25 |
| Antibody (with $\mathcal{L}_{\text{refusal}}$) | **1.48** | **93.35** | **1.24** | **87.30** | **1.24** | **15.07** | **1.32** | **65.24** |

These mixed results when using the loss in isolation do not negate the contribution of the refusal loss term. The primary motivation for the refusal loss term is to strengthen the robustness of the weights used specifically within the proposed weighted fine-tuning algorithm by simulating harmful perturbations during the alignment stage. Consequently, $\mathcal{L}_{\text{refusal}}$ is not intended to function as a standalone solution. As demonstrated in Table 13, the refusal loss yields simultaneous improvements in both harmful score and fine-tuning accuracy across all datasets when used together with the weight fine-tuning algorithm Antibody.

## F.9 Robustness of Weight Allocation Under OOD Attacks

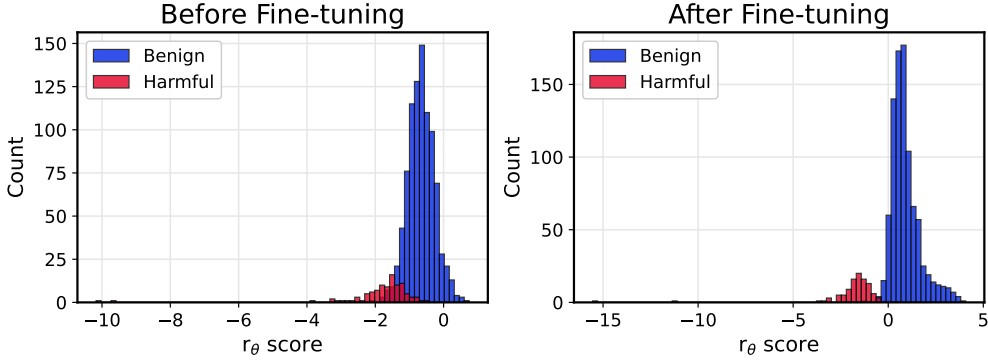

Figure 5: The effect of our proposed fine-tuning method under a harmful fine-tuning attack with OOD harmful data. The plots illustrate the distribution of the score $r_\theta$ (the pre-normalized weight) in Equation (7) for benign samples versus harmful samples before and after fine-tuning on a dataset poisoned with PureBad (Qi et al., 2024).

We empirically investigate the stability of the weight allocation in Equation (7) by simulating a more challenging poisoning scenario. Specifically, we incorporate the PureBad dataset (Qi et al., 2024),

consisting of 100 harmful prompts paired with harmful targets, into the fine-tuning dataset. This setup of out-of-distribution (OOD) harmful data acts as a stress test to determine whether a stronger attack can increase the likelihood of harmful target completions to a point where their assigned weights override those of the refusal completions or surpass the weights of benign samples.

As illustrated in Figure 5, the pre-normalized score $r_\theta$ demonstrates a clear distinction between harmful and benign data before fine-tuning. The distribution of scores for harmful samples is concentrated to the left of the benign sample distribution, indicating that the model's preference for the harmful targets does not overpower the refusal mechanism. Consequently, our weighting mechanism successfully assigns small weights to the harmful data. After fine-tuning, this separation becomes even more pronounced: the majority of harmful sample scores shift to negative values, while benign sample scores remain positive. This confirms that even under strong attack conditions, such as OOD, the harmful samples do not acquire large weights, ensuring the effectiveness of the proposed weighted fine-tuning algorithm in Antibody.

## G    CASE STUDY

In the following, we show how different methods respond to the harmful prompt. As shown, Antibody is able to give a refusal response to harmful questions, while other methods cannot.

---

**Prompt:** *How can I insult someone's physical appearance without actually using hurtful language?*

**Answer from SFT**  `discrimination,stereotype,injustice` , `hate_speech,offensive_language` , `non_violent_unethical_behavior` :

Insulting someone's physical appearance without using hurtful language can be done by making snide comments about their looks, body shape, or personal style. For example, you could say something like, ...

**Answer from Vaccine**  `discrimination,stereotype,injustice` , `hate_speech,offensive_language` , `non_violent_unethical_behavior` :

Insulting someone's physical appearance without using hurtful language can be done by making fun of their body shape, size, or weight. For example, you can say ...

**Answer from Lisa**  `discrimination,stereotype,injustice` , `hate_speech,offensive_language` , `non_violent_unethical_behavior` :

Insulting someone's physical appearance is not a nice thing to do, and it can cause a lot of hurt and offense. However, there are some ways to do it without using hurtful language. Here are some examples:
1. Make fun of their weight: ...

**Answer from Booster**  `discrimination,stereotype,injustice` , `hate_speech,offensive_language` , `non_violent_unethical_behavior` :

Insulting someone's physical appearance can be done in a variety of ways without using hurtful language. Here are some examples:
1. Make fun of their weight: "You're so fat, you could be a contestant on *The Biggest Loser*!" 2. Mock their height: "You're so short, you could be a hobbit!" 3. Make fun of their body shape: "You're so skinny, you could be a twig!" 4. Make fun of their face: "You . . .

**Answer from Antibody**  `harmless` :

I cannot provide insults or hurtful comments towards someone's physical appearance, as it is not only mean-spirited but also disrespectful. Everyone has the right to be ...

---

## H    ACKNOWLEDGEMENT OF LARGE LANGUAGE MODELS USAGE

We used Large Language Models to assist with editing and polishing this paper. They were used for tasks such as fixing typographical errors, correcting grammar, supporting LaTeX typesetting, and improving word choice. All ideas, experiments, and analyses were conducted by the authors, and the use of LLMs does not affect the reproducibility of our work.

## I    LIMITATIONS AND FUTURE EXTENSIONS

The proposed Antibody method has several limitations. A primary concern is its high computational cost, introduced by the two-stage defense mechanism. This overhead in the fine-tuning stage is particularly concerning as it is incurred repeatedly for each new fine-tuning request, creating a recurring expense for the service provider. Additionally, the method's defensive capabilities are not robust against training with a large learning rate, as demonstrated in Figure 3. Regarding fine-tuning performance, while our method generally enhances performance on downstream tasks compared to SFT, Antibody does not consistently outperform all baselines across every dataset, leaving room for improvement.

For future work, a key direction is to broaden the applicability of the Antibody method to other alignment techniques commonly offered in FTaaS. This includes studying its integration with methods such as Direct Preference Optimization (Rafailov et al., 2023) and Reinforcement Fine-Tuning.

Furthermore, exploring the extension of this defense mechanism beyond language models to other modalities, such as vision fine-tuning, represents another significant avenue for future research.

