# OpenReview forum: "Antibody: Strengthening Defense Against Harmful Fine-Tuning for Large Language Models via Attenuating Harmful Gradient Influence"
_ICLR.cc/2026/Conference — ICLR 2026 Poster_

### Official Review · Reviewer_xQyK · 2025-10-26

**Soundness:** 3
**Presentation:** 2
**Contribution:** 3
**Rating:** 6
**Confidence:** 3

**Summary:**

This paper proposes a two-stage defense framework to protect FTaS models from harmful fine-tuning attacks. The first step is regularizing the model on a flat-loss region for harmful samples to make safety alignment more resilient. The second step is the finetuning with weights adjusted for samples based on how harmful they potentially are to further reduce their impacts. Experiments show that Antibody can reduce the harmful score significantly while maintaining accuracy.

**Strengths:**

- The problem is critical and well motivated.
- The idea of optimizing the model to stay in the flat-loss region with respect to harmful samples is interesting.
- Extensive theoretical analysis is provided.
- Experimental results are promising and the ablation studies are extensive.

**Weaknesses:**

- In line 196, authors state that among the models that lie in the flat region of the harmful loss L_harm, they aim to find the one that minimizes the alignment loss L_align. But the optimization objective in Equation 3 is not exactly doing this. Opposite in order, it actually finds the weights that minimize harmful loss among the weights that minimizes alignment loss. I think we need more clarity here.
- It is not clear how we should interpret Theorem 4.1 (which is the most important one in this paper) and what a_t is without looking at the appendix. More explanation is needed in the main body.
- In the second stage, it is assumed that harmful samples are promised to be in the flat-loss region and benign samples won’t be in the flat-loss region, which may not hold for benign samples.
- There is additional computation and pipeline overhead that may significantly increase the time and energy spent during the whole process. Analysis on the computational cost of the introduced first and second stage operations with detailed breakdowns are expected.
- Fig.1 is not displayed correctly (cropped).

**Questions:**

- What do you mean by a test sample in Proposition 4.2? A sample used in finetuning or testing? We are not supposed to compute gradients for a “test” test sample, am I correct?
- How does the size of refusal dataset impact results?

---

> ### Author Response · Authors · 2025-11-21
> **Author Rebuttal (I): Clarification on flatness regularization objective**
>
> We thank the reviewer for their constructive review of our paper. Below are our answers to address the reviewer's questions.
>
> > ### **W1:** In line 196, authors state that among the models that lie in the flat region of the harmful loss L\_harm, they aim to find the one that minimizes the alignment loss L\_align. But the optimization objective in Equation 3 is not exactly doing this. Opposite in order, it actually finds the weights that minimize harmful loss among the weights that minimizes alignment loss. I think we need more clarity here.
>
> We put our line 196 here for reference: "*among the models $\theta$ that lie in the flat region of the harmful loss $\mathcal{L}\_{\text{harm}}$, we aim to find the one that minimizes the alignment loss $\mathcal{L}\_{\text{align}}$.*" Below is our response:
>
> Our objective in Equation (3) does not minimize the harmful loss. The only losses we minimize in Equation (3) are alignment loss $\mathcal{L}\_{\text{align}}$ and sharpness loss $\mathcal{L}\_{\text{sharp}}$. Even though the definition of $\mathcal{L}\_{\text{sharp}}$ involves $\mathcal{L}\_{\text{harm}}$, specifically
>
> $\mathcal{L}\_{\text{sharp}}(\theta') \triangleq \mathcal{L}\_{\text{harm}}(\theta') - \min\_{\phi \in \mathcal{B}\_\rho(\theta')} \mathcal{L}\_{\text{harm}}(\phi),$
>
> it **also includes the subtraction term $- \min\_{\phi\in\mathcal{B}\_{\rho}(\theta')} \mathcal{L}\_{\text{harm}}(\phi)$**. This turns $\mathcal{L}\_{\text{sharp}}$ into a *flatness* measure: it is small when $\mathcal{L}\_{\text{harm}}$ cannot be further reduced within the $\rho$-ball $\mathcal{B}\_\rho(\theta')$, i.e., when the harmful loss is locally flat around $\theta'$.
>
> Additionally, our claim in line 196 is consistent with our objective in Equation (3):
>
> $
> \delta\_t \in \tfrac{1}{2}\operatorname{argmin}\_{\delta} \big\| \nabla\_{\theta}\mathcal{L}\_{\text{align}}(\theta\_t) - \delta \big\|\_{2}^{2}
> \quad \text{s.t.} \quad
> \nabla\_{\theta}\mathcal{L}\_{\text{sharp}}(\theta\_t)^{\top} \delta\_t \ge a\_{t} > 0.
> $
>
> The constraint $\nabla\_{\theta}\mathcal{L}\_{\text{sharp}}(\theta\_t)^{\top}\delta\_t \ge a\_t > 0$ restricts updates to directions that decrease $\mathcal{L}\_{\text{sharp}}$ under $\theta\_{t+1} = \theta\_t - \xi \delta\_t$, thereby moving $\theta\_t$ into (or keeping it within) a flat region of $\mathcal{L}\_{\text{harm}}$. Within this feasible set, the objective on the right of Equation (3) selects the update that best follows the alignment gradient and thus minimizes $\mathcal{L}\_{\text{align}}$. **Hence, we are not minimizing $\mathcal{L}\_{\text{harm}}$ first and then $\mathcal{L}\_{\text{align}}$ in the reverse order. Instead, we minimize $\mathcal{L}\_{\text{align}}$ subject to staying in the flat $\mathcal{L}\_{\text{harm}}$ region.**

---

> ### Author Response · Authors · 2025-11-21
> **Author Rebuttal (II): Clarification about Theorem 4.1 and a_t**
>
> > ### **W2:** It is not clear how we should interpret Theorem 4.1 (which is the most important one in this paper) and what a_t is without looking at the appendix. More explanation is needed in the main body.
>
> We thank the reviewer for pointing this out and for emphasizing the need for a clearer interpretation of Theorem 4.1 and the term $a_t$. We have revised the paper to expand and clarify our discussion of Theorem 4.1 and $a_t$, and we provide a detailed explanation below.
>
> In the constraint $\nabla\_{\theta} \mathcal{L}\_{\text{sharp}}(\theta\_t)^{\top} \delta\_t \ge a\_t > 0$, the scalar $a\_t$ specifies the minimum required decrease of the sharpness loss at step $t$. Using a first-order approximation, $\mathcal{L}\_{\text{sharp}}(\theta\_{t+1}) - \mathcal{L}\_{\text{sharp}}(\theta\_t) \approx -\xi \, \nabla\_{\theta} \mathcal{L}\_{\text{sharp}}(\theta\_t)^{\top} \delta\_t \le -\xi a\_t < 0$ for a sufficiently small step size $\xi$, any choice of $a\_t > 0$ guarantees that $\mathcal{L}\_{\text{sharp}}$ strictly decreases at each iteration, i.e., each update moves the model further into a flat harmful loss region. In practice, we set $a\_t = \xi \|\nabla\_{\theta} \mathcal{L}\_{\text{sharp}}(\theta\_t)\|\_2^2$, which yields a simple dynamic rule: when the sharpness gradient is large, the constraint is stricter (more aggressive flattening), and when it is small, the constraint is mild and the update behaves closer to standard alignment.
>
> Theorem 4.1 then shows that the optimal update direction solving Eq. (7) is $\delta\_t^{*} = \nabla\_{\theta} \mathcal{L}\_{\text{align}}(\theta\_t) + \lambda\_t \nabla\_{\theta} \mathcal{L}\_{\text{sharp}}(\theta\_t)$, and the explicit formula of $\lambda\_t$. Intuitively, if the alignment gradient $\mathcal{L}\_{\text{align}}$ already makes sufficient progress in reducing sharpness, we have $\lambda\_t = 0$ and recover a pure alignment step. Otherwise, $\lambda\_t > 0$ adds just enough of the sharpness gradient to ensure the required decrease in $\mathcal{L}\_{\text{sharp}}$, thereby jointly optimizing alignment and flat harmful loss.

---

> ### Author Response · Authors · 2025-11-21
> **Author Rebuttal (III): Clarification about the proposed fine-tuning stage defense**
>
> > ### **W3:** In the second stage, it is assumed that harmful samples are promised to be in the flat-loss region and benign samples won’t be in the flat-loss region, which may not hold for benign samples.
>
> The benign samples are not in the flat-loss region. In our alignment stage, the sharpness loss $\mathcal{L}_{\text{sharp}}$ is computed only on harmful data, and we explicitly control the loss sharpness with respect to harmful samples. No benign samples (e.g., math-related examples) are included in the safety alignment process, so we do not impose any sharpness constraints on benign losses.
>
> Empirically, benign samples do not appear to be pushed into a flat-loss region: the solid blue curve in Figure 2 (benign loss of our method) shows a clear decreasing trend during the fine-tuning stage. Furthermore, the downstream task results in the Experiments section show strong benign-task performance, indicating that the loss on benign samples decreases in the fine-tuning stage rather than remaining at a stable value due to the sharpness regularization on harmful data. Finally, we have added Figure 4 to Appendix F.6 that demonstrates the effect of our flatness regularization in the alignment stage and validates the claim that benign samples are not in a flat region. Figure 4 in Appendix F.6 demonstrates that, for the model aligned with Antibody, the distribution of gradient norms for benign samples lies to the right of the harmful samples. These norms are clearly larger than those of the harmful samples and not negligible, indicating that the benign samples are not in a flat loss region.
>
> > ### **Q1:** What do you mean by a test sample in Proposition 4.2? A sample used in finetuning or testing? We are not supposed to compute gradients for a “test” test sample, am I correct?
>
> You are absolutely correct that we are **not supposed to compute gradients for any “test” test sample**. In the context of Proposition 4.2, the test sample $(\mathbf{x}_o, \mathbf{y}_o)$ refers to any arbitrary sample whose loss change we wish to analyze *hypothetically* as a result of a mini-batch update on other training samples.
>
>
> The primary purpose of introducing this theoretical test sample is to analyze the *dynamics* of learning. For a test sample $(\mathbf{x}_o, \mathbf{y}_o)$ that belongs to the same task as the benign sample $(\mathbf{x}_i, \mathbf{y}_i)$ (i.e., $\lVert K^t \rVert_F$ is large), the mini-batch update will decrease the loss on this test sample. In contrast, for harmful test samples that are inherently dissimilar to the benign sample in the batch (small $\lVert K^t \rVert_F$), this update will not change the loss on $(\mathbf{x}_o, \mathbf{y}_o)$, thus preserving the model’s alignment.

---

> ### Author Response · Authors · 2025-11-21
> **Author Rebuttal (IV): Experiment on computational overhead**
>
> > ### **W4:** There is additional computation and pipeline overhead that may significantly increase the time and energy spent during the whole process. Analysis on the computational cost of the introduced first and second stage operations with detailed breakdowns are expected.
>
> We have already conducted the system analysis in the original submission of the paper (Appendix F.4). We reiterate the results in the following table for completeness and to address the reviewer's concern.
>
> | Methods (Llama-2-7B) |  | **Clock Time (Hours)** |  |  |  | **GPU Memory (GB)** |  |
> |:---|:---:|:---:|:---:|:---:|:---:|:---:|:---:|
> | | Alignment | Fine-tuning | Sum |  | Alignment | Fine-tuning | Max |
> | SFT | 1.00 | 0.26 | 1.26 |  | 53.68 | 40.10 | 53.68 |
> | Vaccine | 1.98 | 0.16 | 2.12 |  | 54.70 | 40.10 | 54.70 |
> | Lisa | 1.00 | 0.26 | 1.26 |  | 53.68 | 53.87 | 53.87 |
> | Booster | 5.09 | 0.26 | 5.35 |  | 64.36 | 40.10 | 64.36 |
> | Antibody | 5.81 | 0.46 | 6.27 |  | 64.55 | 42.53 | 64.55 |
>
> The table presents the system evaluation results, which illustrate that the comprehensive protection offered by our proposed Antibody method comes at the cost of some additional computational resources. During the alignment stage, Antibody requires 5.81 hours and 64.55 GB of GPU memory, higher than other methods. This overhead is attributed to Antibody's sophisticated alignment process, which not only imposes an initial safeguard but also strategically shapes the model's loss landscape for the subsequent fine-tuning defense. Unlike methods that operate solely during alignment, Antibody also introduces a defense mechanism in the fine-tuning stage. This results in a marginal increase in clock time (0.20 hours) and GPU memory (2.43 GB) compared to standard fine-tuning. **We argue this is a strategic trade-off: the *one-time alignment cost* builds a robust foundation, while the *minimal*, *recurring overhead* in the fine-tuning stage acts as an active safeguard for each downstream task.** While its total computational cost is the highest, this investment is justified by its superior defense and learning performance. The minimal overhead during the frequently repeated fine-tuning phase makes Antibody a practical and highly effective defense solution for deploying in FTaaS scenarios.

---

> ### Author Response · Authors · 2025-11-21
> **Author Rebuttal (V): Experiment on refusal dataset size**
>
> > ### **Q2:** How does the size of refusal dataset impact results?
>
> First, we want to clarify that the dataset used in the alignment stage of Antibody include $\mathcal{D}\_{\text{align}}$, $\mathcal{D}\_{\text{harmful}}$, and $\mathcal{D}\_{\text{refusal}}$. The number of samples in these three datasets is equal by construction, as detailed below:
>
> - $\mathcal{D}\_{\text{align}}$ consists of harmful prompt–refusal completion pairs.
> - $\mathcal{D}\_{\text{harmful}}$ consists of harmful prompt–compliance answer pairs, where the harmful prompts are the exact same prompts as in $\mathcal{D}\_{\text{align}}$. Thereby, the size of $\mathcal{D}\_{\text{harmful}}$ and $\mathcal{D}\_{\text{align}}$ are equal.
> - $\mathcal{D}\_{\text{refusal}}$ consists of harmful prompt pairs with generic refusal responses that contain refusal answers that can be used across different harmful prompts, such as "I cannot fulfill your request." Hence, the size of $\mathcal{D}\_{\text{refusal}}$ and $\mathcal{D}\_{\text{align}}$ are the same.
>
> In the following, we refer to the common size of $\mathcal{D}\_{\text{align}}$, $\mathcal{D}\_{\text{harmful}}$, and $\mathcal{D}\_{\text{refusal}}$ as the *alignment size*. **We conduct experiments to study how the defense and learning performance of Antibody and Booster change with different alignment sizes.**
>
> Table: Performance comparison of Booster and Antibody methods across different alignment sizes. The best results are highlighted in bold. Note that 4972 corresponds to the alignment dataset size used in our main experiments, and Booster uses $\mathcal{D}\_{\text{align}}$ and $\mathcal{D}\_{\text{harmful}}$ in its algorithm.
>
> | Alignment Size | Method   | SST2 HS $\downarrow$ | SST2 FA $\uparrow$ | AGNEWS HS $\downarrow$ | AGNEWS FA $\uparrow$ | GSM8K HS $\downarrow$ | GSM8K FA $\uparrow$ | Avg HS $\downarrow$ | Avg FA $\uparrow$ |
> |---------------:|----------|----------------------|--------------------|------------------------|----------------------|-----------------------|---------------------|---------------------|-------------------|
> | 1000           | Booster  | 34.05                | 93.46              | 40.49                  | 86.30                | 43.49                 | **15.80**           | 39.34               | **65.19**         |
> | 1000           | Antibody | **5.44**             | **93.96**          | **5.29**               | **86.40**            | **8.44**              | 14.10               | **6.39**            | 64.82             |
> | 2500           | Booster  | 22.46                | 92.66              | 21.32                  | 85.10                | 31.19                 | 15.10               | 24.99               | 64.29             |
> | 2500           | Antibody | **2.00**             | **93.81**          | **2.15**               | **86.90**            | **6.44**              | **16.10**           | **3.53**            | **65.60**         |
> | 4972           | Booster  | 14.31                | 92.59              | 15.88                  | 86.70                | 9.06                  | **16.27**           | 13.08               | 65.19             |
> | 4972           | Antibody | **1.48**             | **93.55**          | **1.24**               | **87.30**            | **1.24**              | 15.07               | **1.32**            | **65.31**         |
>
> From this, a notable trend is that both methods benefit from increased alignment size, but Antibody scales more effectively. Across all settings and datasets, Antibody consistently achieves a much lower harmful score than Booster while maintaining comparable or slightly better fine-tuning accuracy. With an alignment size of 1000, Antibody reduces the average harmful score from 39.34 for Booster to 6.39, more than a six-fold reduction, while maintaining comparable fine-tuning accuracy (64.82 for Antibody and 65.19 for Booster). Increasing the alignment size to 2500 further strengthens this effect: Antibody attains an average harmful score of 3.53, whereas Booster remains at 24.99, and Antibody slightly improves the average fine-tuning accuracy to 65.60 compared to 64.29 for Booster. At the largest alignment size of 4972, Antibody simultaneously reduces the average harmful score to 1.32 (compared with Booster's 13.08) and increases the average fine-tuning accuracy to 65.31 (slightly above Booster's 65.19). **Overall, these results indicate that, for a given alignment size, Antibody provides substantially stronger and more data-efficient protection against harmful fine-tuning than Booster, even when only 1000 alignment examples are available.**

---

> ### Author Response · Authors · 2025-11-21
> **Author Rebuttal (VI): Figure 1 does not display correctly on Safari's PDF viewer**
>
> > ### **W5:** Fig.1 is not displayed correctly (cropped).
>
> We thank the reviewer for reporting this issue. We have identified that Figure 1 may not render correctly in Safari's PDF viewer. The figure displays properly in Chrome, Firefox, or when downloaded and viewed locally. We recommend using one of these alternatives.

---

> ### Comment · Reviewer_xQyK · 2025-11-26
>
> Thanks for the detailed and well-organized rebuttal. My concerns are mostly addressed. I will revise my rating accordingly.

---

> ### Author Response · Authors · 2025-11-26
> **Thank the reviewer for the update**
>
> We thank the reviewer for the follow-up. We are glad to hear that our rebuttal successfully addressed your concerns, and we appreciate your decision to revise the rating.

---

### Official Review · Reviewer_jv5a · 2025-10-27

**Soundness:** 4
**Presentation:** 2
**Contribution:** 4
**Rating:** 8
**Confidence:** 5

**Summary:**

This paper proposes an elegant solution towards harmful fine-tuning. The solution contains two main components, an alignment stage solution and a fine-tuning stage solution. The design is elegant and with sufficient ablation study to justify each components usefulness.  Experiments show that it outperforms the SOTA solution Booster.

**Strengths:**

1. The studied problem is important and this papers offer a timely contribution.
2. The solution is elegant and very fundamental. I especially like the alignment stage component of Antibody, as it enhances safety alignment by adding a few regularizer in the safety alignment stage.
3. Amplified experimental results are given, showing the robustness of such solutions on different fine-tuning datasets, model and harmful ratios. Also, I particularly like Figure 1, **which clearly demonstrates  Antidote's superiority against existing methods over the trade-off between safety and fine-tune accuracy**.
4. The authors conduct sufficient ablation study,  system evaluations, and quantitative results to verify the effectiveness of the proposed method.
5. The visualization in Figure 2 is nice!


In conclusion, this is a solid paper that requires a lot of experiment and sufficient insights and understanding of the problem. I feel pleased and super excited when reading this paper.

**Weaknesses:**

The paper contains two components, I.e., alignment stage and fine-tuning stage solution. I need to separate them to write the review for each of them for clarity.

For alignment stage of Antibody:

1. **Relation and contribution compared with Booster need to be clarified.**
The relation with Booster is not explicitly mentioned. Particularly, the authors should clarify that Antibody inherits components from Booster. Let's look at the update rule in  Eq. (11).

* **Under the case $\lambda_{refusal}=0$ and $\lambda_t = \lambda$ (i.e., a constant), Antibody reduces to Booster**.

Justification: under the above assumption  on \lambda_{refusal} and  $\lambda_t $,  then, expands $L_{sharp}(\theta)$ based on its definition given in line 222 in Page 5, Eq.  (11) becomes:

$L_{align}(\theta_t) + \lambda L_{harm}(\theta_t) - L_{harm}(\theta_t- \rho \frac{ \nabla L_{harm}(\theta)) }{||\nabla L_{harm}(\theta)) || } ) $

**This update rule is 100\% the same with Booster's update rule. See Eq. (1) in Booster[1].**

[1] Booster: Tackling Harmful Fine-tuning for Large Language Models via Attenuating Harmful Perturbation. (ICLR2025)

If the authors  agree that the two methods share similarity at some level, I recommend the authors to clarify this relation at the end of Section 4.1, such that non-experts in this sub-field could better understand what the real contribution is. **Also, given that the final derived update rule of solving Equation (2) is almost identical to Booster, could you clarify what the contribution of Section 4.1 is?**  Feel free to challenge me in the rebuttal if you do not agree with what I claim.


2. **The real contribution seems to be the third term in Eq. (11), which is not clearly articulated**.  As I mentioned above, Section 3.1 (which introduces first two terms in Eq, (11)) arrives at an update rule that is almost identical to Booster. However, by looking at Eq. (11) again, actually Antibody contains an extra term that is not considered by Booster. Particularly the $\lambda_{refusal} L_{refusal}(\theta_{pert,t})$ is a new term that is not considered by existing work. To my understanding, this term is important as it makes sure that the safety alignment loss will not increase after simulating one step of harmful perturbation. That might be the key contribution that guarantee Antiboidy' superior performance against Booster.

To verify the meaningfulness of this contribution, I suggest the authors to do a group of ablation experiments:

* Disable the fine-tuning stage solution of Antibody, and only keep the alignment part.  Then, set $\lambda_t$ to a constant, making the first two terms reduce to Booster. Finally, replicating a similar experiment settings in figure 1 by comparing $\lambda_{refusal}=0$ (reduces to Booster)  and   $\lambda_{refusal}$=an ideal number.

3. **While they are not exactly the same, a related alignment stage harmful fine-tuning defense paper also uses a similar term with the third term in Eq. (11).** Particularly, the second term in  Eq. (2) in [2] also adopt a similar term, which can be adapted to $\lambda_{collapse} L_{collpase}(\theta_{pert,t})$ based on the notation used by this paper. The difference is in the dataset used for the outer loss calculation.  Antibody is on the alignment dataset and CTRAP is on a specially constructed collapse dataset. I suggest the authors add a sentence to discuss this term in CTRAP paper to mention their similarity.

[2] CTRAP: Embedding Collapse Trap to Safeguard Large Language Models from Harmful Fine-Tuning

For fine-tuning stage of Antibody:

4. **Relation and contribution compared with SEAL[3] need to be clarified.** The idea of the fine-tuning stage solution is to assign different weights for each piece of data in the fine-tuning datasets. The high level idea is similar  to SEAL which assign a learnable weights to each data point. The authors should discuss how the assigned weights are different from SEAL and justify with experiment whether this weight allocation in Eq. (7) are better than the learnable weights in SEAL [3]. Also, there is one NeurIPS2025 accepted paper BDS[4] and two pre-print papers GradShield[5] and Ref-Teacher[6] exploring the same direction, which should also be mentioned in the paper.

[3] SEAL: Safety-enhanced Aligned LLM Fine-tuning via Bilevel Data Selection

[4] Adaptive Defense against Harmful Fine-Tuning via Bayesian Data Scheduler

[5]  GradShield: Alignment Preserving Finetuning  https://openreview.net/forum?id=YYUNm7IibC

[6] Safety-Aligned Weights Are Not Enough: Refusal-Teacher-Guided Finetuning Enhances Safety and Downstream Performance under Harmful Finetuning Attacks  https://openreview.net/forum?id=OK2GR1guwv



5. I have some concerns over the  weights allocation in (7).  I think this weight allocation might incur some issues when the model is under stronger attack. If the attack is strong, the likelihood of the harmful samples of generating the harmful target completion might override that of the refusal completion. In that case. the weight allocation towards harmful samples might becomes higher and even might surpass that of the benign sample. Could you give me some insights on whether this case will happen?

6. **Some related work on harmful fine-tuning defense should be discussed.**

Detecting Adversarial Fine-tuning with Auditing Agents

Scaling Trends for Data Poisoning in LLMs

Unleashing the Unseen: Harnessing Benign Datasets for Jailbreaking Large Language Models

Virus: Harmful Fine-tuning Attack for Large Language Models Bypassing Guardrail Moderation

No, of course I can! Refusal Mechanisms Can Be Exploited Using Harmless Fine-Tuning Data

Benign Samples Matter! Fine-tuning On Outlier Benign Samples Severely Breaks Safety

Your Agent May Misevolve: Emergent Risks in Self-evolving LLM Agents

Eliciting Harmful Capabilities by Fine-Tuning on Safeguarded Outputs

Deep Ignorance: Filtering Pretraining Data Builds Tamper-Resistant Safeguards into Open-Weight LLMs

Self-Destructive Language Model

CTRAP: Embedding Collapse Trap to Safeguard Large Language Models from Harmful Fine-Tuning

Vulnerability-Aware Alignment: Mitigating Uneven Forgetting in Harmful Fine-Tuning

LoX: Low-Rank Extrapolation Robustifies LLM Safety Against Fine-tuning

Towards Resilient Safety-driven Unlearning for Diffusion Models against Downstream Fine-tuning

SEAL: Safety-enhanced Aligned LLM Fine-tuning via Bilevel Data Selection

Safety alignment should be made more than just a few tokens deep

Beware of Your Po! Measuring and Mitigating AI Safety Risks in Role-Play Fine-Tuning of LLMs

Shape it Up! Restoring LLM Safety during Finetuning

Mitigating Fine-tuning Risks in LLMs via Safety-Aware Probing Optimization

Refusal-Feature-guided Teacher for Safe Finetuning via Data Filtering and Alignment Distillation

AsFT: Anchoring Safety During LLM Fine-Tuning Within Narrow Safety Basin

Your Task May Vary: A Systematic Understanding of Alignment and Safety Degradation when Fine-tuning LLMs

Defending MoE LLMs against Harmful Fine-Tuning via Safety Routing Alignment

GradShield: Alignment Preserving Finetuning

A Guardrail for Safety Preservation: When Safety-Sensitive Subspace Meets Harmful-Resistant Null-Space

Detecting Instruction Fine-tuning Attack on Language Models with Influence Function

Locking Down the Finetuned LLMs Safety

Panacea: Mitigating Harmful Fine-tuning for Large Language Models via Post-fine-tuning Perturbation

Safe Delta: Consistently Preserving Safety when Fine-Tuning LLMs on Diverse Datasets

Navigating the safety landscape: Measuring risks in finetuning large language models

Fundamental Safety-Capability Trade-offs in Fine-tuning Large Language Models

ESTIMATING WORST-CASE FRONTIER RISKS OF OPEN-WEIGHT LLMS

When Style Breaks Safety: Defending Language Models Against Superficial Style Alignment

**There may be more relevant works (I just list above some more recent work), and I suggest the authors to read and discuss all of the relevant works on harmful fine-tuning when revising the paper.**

**Questions:**

See Weakness. Overall I enjoy reading this paper, and I will consider to further improve my score if my concerns are addressed.

The main concern I have is that this paper should not emphasize too much on the flatness regularization as this is exactly the same with what Booster aim to achieve (also evidenced by the almost identical update rule between Antibody and Booster), though it seems that the constraint problem Eq. (2) offers an alternative explanation of the flatness regularization and also by this transformation you seem to be able to automatically and adaptively derive the regularizer intensity $\lambda_t$, which seems to be a novel contribution. You may talk more about this if you agree to mention Booster formulation in this section.  **I also suggest to add a group of experiments to justify that the step-adaptive regularizer intensity is indeed necessary and outperform the constant intensity adopted by Booster**, if you still insist that your Section 4.1  should be the main contribution.

The third term in (11) (i.e.,the term in Eq. (10) is more interesting and I personally think is the main contribution instead.

Despite I do have some  complaint over the writing issue, I think this paper is a very solid work and I am happy to champion this paper in the review process.

---

> ### Author Response · Authors · 2025-11-21
> **Author Rebuttal (I): Connection with Booster**
>
> We sincerely thank the reviewer for their insightful comments and for identifying connections between Antibody and related methods, especially Booster [1]. We are happy to provide our detailed discussion below.
>
> ### **W1 + Q1: Clarification on Relation to Booster and Contributions of Section 4.1**
>
> In the following, we clarify the relationship between Booster and our alignment-stage defense method in Section 4.1.
>
> **1. Relation to Booster and Convergence of Update Rules.**
> We fully agree with your observation. If we set $\lambda_{\text{refusal}} = 0$ and fix the adaptive weight $\lambda_t$ to a constant scalar, the update rule derived from our sharpness minimization term indeed converges to the formulation used in Booster.
>
> **2. Motivation & Algorithmic Contributions of Section 4.1.**
> While the resulting update rules share similarities in that specific limit, our flatness regularized method distinguishes itself through its **motivation** and the **adaptive regularizer intensity**:
>
> *   **Motivation:** Booster aims to minimize the reduction of harmful loss during the fine-tuning stage by simulating harmful fine-tuning via harmful perturbations in the alignment stage. This is implemented by minimizing the gap in harmful loss before and after the perturbation. In contrast, our method is derived from a constrained optimization problem aimed at seeking a *flat loss landscape* with respect to harmful samples. This forces the model into a region where fine-tuning attacks are ineffective due to smaller harmful gradients (as detailed in the newly added Appendix F.6). Our definition of $\mathcal{L}_{\text{sharp}}$ as the difference between the current harmful loss and the local minimum leads to a formulation that mathematically resembles the loss term in Booster when a single gradient descent step is used to find the local minimum. This relationship offers an additional interpretation of Booster's proposed loss term.
> *   **Adaptive regularizer intensity $\lambda_t$:** A key contribution of Section 4.1, as you noted, is the derivation of the adaptive regularizer intensity $\lambda_t$. In Booster, the coefficient is a fixed hyperparameter. Our formulation (Theorem 4.1) dynamically calculates $\lambda_t$ at every step to satisfy the descent constraints.
>
> **We have revised the paper to explicitly clarify this relationship at the end of Section 4.1.**
>
> **3. The refusal loss term $\mathcal{L}_{\text{refusal}}$ in Equation (10/11).**
> We agree with your assessment that the Refusal Locking term is a vital contribution. However, we consider both the flatness regularization in Section 4.1 and the refusal loss term to be significant and complementary contributions: while flatness reduces the *magnitude* of harmful updates, $\mathcal{L}_{\text{refusal}}$ ensures the weighting mechanism remains *robust* against parameter drift during fine-tuning. Together, they form a more effective defense than either component in isolation.
>
> Following the reviewer's suggestion, we conduct the ablation study investigating the impact of the step-wise adaptive regularizer $\lambda_t$. We compare the performance of both Booster and Antibody when using a fixed constant regularization versus our proposed dynamic $\lambda_t$. The best results are highlighted in bold.
>
> | Method                               | SST2 HS  | SST2 FA   | AGNEWS HS | AGNEWS FA | GSM8K HS | GSM8K FA  | Avg HS | Avg FA |
> | ------------------------------------ | -------- | --------- | --------- | --------- | -------- | --------- | ------ | ------ |
> | Ablation on Booster                  |          |           |           |           |          |           |        |        |
> | Booster (with constant $\lambda$)    | 14.31    | **92.59** | 15.88     | **86.70** | **9.06** | **16.27** | 13.08  | **65.19** |
> | Booster (with dynamic $\lambda_t$)   | **8.20** | 92.55     | **8.97**  | 85.93     | 14.93    | 15.97     | **10.70** | 64.82  |
> |                                      |          |           |           |           |          |           |        |        |
> | Ablation on Antibody                 |          |           |           |           |          |           |        |        |
> | Antibody (with constant $\lambda$)   | 1.91     | 93.24     | 1.62      | **88.17** | 2.15     | **15.60** | 1.89   | **65.67** |
> | Antibody (with dynamic $\lambda_t$)  | **1.48** | **93.35** | **1.24**  | 87.30     | **1.24** | 15.07     | **1.32** | 65.24  |
>
> We observe that the proposed dynamic $\lambda_t$ consistently reduces the harmful score across both Booster and Antibody, albeit with a slight trade-off in fine-tuning accuracy. Notably, this safety improvement is less pronounced in Antibody. We attribute this to the fact that our proposed weighted SFT mechanism already effectively suppresses the influence of harmful samples, thereby significantly lowering the baseline harmful score and leaving less margin for further reduction by the adaptive regularizer.

---

> ### Author Response · Authors · 2025-11-21
> **Author Rebuttal (II): More experiments on the refusal loss term**
>
> ### **W2: Study the contribution of the refusal loss term $\mathcal{L}_{\text{refusal}}$ in isolation**
>
> We have revised the paper and added Appendix F.8 to study the contribution of the refusal loss term $\mathcal{L}_{\text{refusal}}$.
>
> **Table 12.** Performance under different numbers of fine-tuning samples in the default setting. The best results are highlighted in bold.
> | Methods **(p = 0.2)** | Harmful Score ↓ | | | | | Fine-tuning Accuracy ↑ | | | | |
> |---|---:|---:|---:|---:|---:|---:|---:|---:|---:|---:|
> | | n=500 | n=1000 | n=1500 | n=2000 | n=2500 | n=500 | n=1000 | n=1500 | n=2000 | n=2500 |
> | Booster | 1.62 | 9.06 | 31.76 | **39.49** | **44.49** | **13.30** | **16.27** | **18.27** | 18.83 | 19.90 |
> | Booster + $\mathcal{L}_{\text{refusal}}$ | **1.71** | **5.34** | **28.90** | 41.82 | 48.45 | 11.90 | 15.33 | 16.47 | **18.87** | **20.23** |
>
> This table (Table 12 in Appendix F.8) presents the performance of the model under varying numbers of fine-tuning samples. We utilize the Booster baseline for this comparison, as it lacks the specific fine-tuning stage solution found in our proposed method. This allows us to isolate and analyze the independent effect of the refusal loss term. The results indicate that $\mathcal{L}_{\text{refusal}}$ effectively reduces the harmful score when the number of fine-tuning samples is low to moderate (e.g., $n=1000$ and $n=1500$). However, regarding utility, we observe a trade-off: the refusal loss tends to slightly reduce fine-tuning accuracy in low-sample regimes while boosting accuracy as the sample size increases ($n \ge 2000$).
>
> **Table 13.** Ablation on Antibody with/without $\mathcal{L}_{\text{refusal}}$ across different datasets.
>
> | **Method & Ablation** | **SST2 HS ↓** | **SST2 FA ↑** | **AGNEWS HS ↓** | **AGNEWS FA ↑** | **GSM8K HS ↓** | **GSM8K FA ↑** | **Average HS ↓** | **Average FA ↑** |
> |---|---:|---:|---:|---:|---:|---:|---:|---:|
> | Antibody (without $\mathcal{L}_{\text{refusal}}$) | 3.58 | 92.78 | 2.53 | 85.70 | 5.06 | 14.27 | 3.72 | 64.25 |
> | Antibody (with $\mathcal{L}_{\text{refusal}}$) | **1.48** | **93.35** | **1.24** | **87.30** | **1.24** | **15.07** | **1.32** | **65.24** |
>
> These mixed results when using the loss in isolation do not negate the contribution of the refusal loss term. The primary motivation for the refusal loss term is to strengthen the robustness of the weights used specifically within the proposed weighted fine-tuning algorithm by simulating harmful perturbations during the alignment stage. Consequently, $\mathcal{L}_{\text{refusal}}$ is not intended to function as a standalone solution. As demonstrated in Table 13, **the refusal loss yields simultaneous improvements in both harmful score and fine-tuning accuracy across all datasets when used together with the weighted fine-tuning algorithm in Antibody.**

---

> ### Author Response · Authors · 2025-11-21
> **Author Rebuttal (III): Discussion of Related Work**
>
> We thank the reviewer for pointing out the need to clarify the connection of our method with other works. In the following, we provide a detailed comparison of our method with CTRAP [2], SEAL [3], and mention other recent works in the revised version of the paper.
>
> ### **W3: Connection with CTRAP**
>
> We have added a section in Appendix B to discuss the connection between our third term in Eq. (11) and the corresponding term in CTRAP.
>
> [2] CTRAP: Embedding Collapse Trap to Safeguard Large Language Models from Harmful Fine-Tuning
>
> ### **W4: Connection with SEAL**
>
> We have revised our paper with a discussion of the relationship and contributions between Antibody and SEAL in Appendix C. Our detailed responses are provided below.
>
> SEAL proposes a bilevel data-selection framework. It trains a separate data selector via bilevel optimization using an alignment dataset and a target user fine-tuning dataset, then filters that dataset (keeping the top-$p\%$ samples under the learned ranking) before running SFT on the filtered subset. Although the weights computed from the data selector in SEAL and the weighting mechanism in Antibody aim to reduce the influence of harmful fine-tuning data, they have different practical usages and differ substantially in mechanism.
>
> - **Different in usage settings.** SEAL requires training a dataset-specific selector because its bilevel objective couples the selector to the target fine-tuning set. This is well-suited to settings where the target dataset is known and reused, but may not be favorable in FTaaS, where service providers may face many heterogeneous, one-off user-submitted datasets. In such a service setting, SEAL introduces additional overhead to (i) train a selector per user dataset and (ii) store the selector for future use and If the selector is discarded after a single run, this cost is largely wasted. In contrast, our defense is embedded directly into the fine-tuning process and is immediately deployable to arbitrary user datasets without training any extra model as the data selector. Our method's incurred overhead is studied in Appendix F.4.
> - **Robust with different attack strengths.** SEAL requires choosing a cutoff $p\%$ to decide which samples are retained. This hard threshold can be sensitive to unknown attack strength (e.g., varying poisoning ratios), and a fixed $p$ may under-filter in stronger attacks or over-filter in benign-heavy datasets. Our method avoids any discrete cutoff: all samples remain in the batch, but harmful ones are down-weighted.
> - **Static ranking vs. dynamic reweighting.** The weights in SEAL are used to produce a global ranking for all samples in the fine-tuning dataset for filtering. Once the selector is trained, the ranking is fixed throughout SFT. Our weights serve a different role: they *directly modulate gradient contributions* to encourage learning from benign samples and suppress learning from harmful ones. Moreover, our weights are computed exclusively for the samples in the active mini-batch and are recalculated at every iteration of fine-tuning, allowing them to adapt to the evolving model state (e.g., as the model becomes a better fit on benign data, the weights for benign data become larger). This online, model-dependent reweighting is fundamentally different from SEAL's offline, fixed ranking.
>
> In summary, SEAL's bilevel formulation provides a clean and interpretable way to cast harmful fine-tuning defense as data selection. However, for the dynamic nature of FTaaS, where data distributions vary unpredictably and efficiency is critical, our weighting mechanism provides different usage settings and properties. By utilizing online, model-dependent reweighting rather than static offline filtering, our approach offers a more robust and lightweight solution that eliminates the need for auxiliary training and adaptive hyperparameter tuning. **We believe that this discussion has shown the differences and benefits of our weighting mechanism over SEAL.**
>
> Finally, other works like BDS [4], GradShield [5], and Ref-Teacher [6] have been added to the extended Related Work in the Appendix.
>
> > ### **W6: Some related work on harmful fine-tuning defense should be discussed.**
>
> We have added an extended discussion of related work in Appendix A.

---

> ### Author Response · Authors · 2025-11-21
> **Author Rebuttal (IV): Robustness of Weight allocation under strong attacks**
>
> > ### **W5: Robustness of Weight allocation under strong attacks**
>
> We have investigated the stability of the weight allocation in Equation (7) by simulating a more challenging poisoning scenario, and the results are updated in Appendix F.9. Specifically, we incorporate the PureBad dataset [7], consisting of 100 harmful prompts paired with harmful targets, into the fine-tuning dataset. This setup of out-of-distribution (OOD) harmful data acts as a stress test to determine whether a stronger attack can increase the likelihood of harmful target completions to a point where their assigned weights override those of the refusal completions or surpass the weights of benign samples.
>
> As illustrated in Figure 5 (Appendix F.9), the pre-normalized score $r_\theta$ demonstrates a clear distinction between harmful and benign data before fine-tuning. The distribution of scores for harmful samples is concentrated to the left of the benign sample distribution, indicating that the model's preference for the harmful targets does not overpower the refusal mechanism. Consequently, our weighting mechanism successfully assigns small weights to the harmful data. After fine-tuning, this separation becomes even more pronounced: the majority of harmful sample scores shift to negative values, while benign sample scores remain positive. **This confirms that even under strong attack conditions, such as OOD, the harmful samples do not acquire large weights, ensuring the effectiveness of the proposed weighted fine-tuning algorithm in Antibody.**
>
> ---
> **References:**
>
> [1] Booster: Tackling Harmful Fine-tuning for Large Language Models via Attenuating Harmful Perturbation. (ICLR2025)
>
> [2] CTRAP: Embedding Collapse Trap to Safeguard Large Language Models from Harmful Fine-Tuning
>
> [3] SEAL: Safety-enhanced Aligned LLM Fine-tuning via Bilevel Data Selection. (ICLR 2025)
>
> [4] Adaptive Defense against Harmful Fine-Tuning for Large Language Models via Bayesian Data Scheduler. (NeurIPS 2025)
>
> [5] GradShield: Alignment Preserving Finetuning https://openreview.net/forum?id=YYUNm7IibC
>
> [6] Safety-Aligned Weights Are Not Enough: Refusal-Teacher-Guided Finetuning Enhances Safety and Downstream Performance under Harmful Finetuning Attacks https://openreview.net/forum?id=OK2GR1guwv
>
> [7] Fine-tuning Aligned Language Models Compromises Safety, Even When Users Do Not Intend To! (ICLR 2024)

---

> ### Comment · Reviewer_jv5a · 2025-11-23
> **Thanks for the rebuttal**
>
> For table 1 in the rebuttal, here the differerence between "Ablation on Booster" and "Ablation on Antibody" is that Antibody contains the second data selection component while Booster does not contain the data selection component? Is my understanding correct?
>
>  It seems that for GSM8k adding the dyncamic regularizer to Booster does not help with better safety performance (9.06 vs 14.93). Could you try some other random seeds to see what happen to this task? With that said, it seems that for Antibody (which add the data selection), the safety performance is consistently better.
>
> I think I will keep the  acceptance score for now.

---

> ### Author Response · Authors · 2025-11-24
> **Thank the reviewer for the follow-up questions**
>
> In Table 1 of the rebuttal, the difference between “Ablation on Booster” and “Ablation on Antibody” is that Antibody includes **(i) the refusal loss term $\mathcal{L}_{\text{refusal}}$ during the alignment stage and (ii) our safety-weighted fine-tuning algorithm during the fine-tuning stage.**
>
> All of our experiments were conducted using three random seeds. Below, we provide the per-seed breakdown not only for GSM8K but also for SST2 and AGNEWS:
>
>
> | Different seed                     | SST2 HS   | SST2 FA   | AGNEWS HS | AGNEWS FA | GSM8K HS | GSM8K FA  |
> | ---------------------------------- | --------- | --------- | --------- | --------- | -------- | --------- |
> | seed=5                             |           |           |           |           |          |           |
> | Booster (with constant $\lambda$)  | 6.15      | **92.55** | 10.01     | **85.80** | 18.03    | 17.20     |
> | Booster (with dynamic $\lambda_t$) | **5.15**  | 92.43     | **6.87**  | 83.40     | **8.73** | **17.60** |
> |                                    |           |           |           |           |          |           |
> | seed=6                             |           |           |           |           |          |           |
> | Booster (with constant $\lambda$)  | 20.46     | **92.78** | 17.17     | 87.20     | **3.58** | **15.20** |
> | Booster (with dynamic $\lambda_t$) | **6.44**  | 92.43     | **6.87**  | **87.30** | 22.32    | 14.70     |
> |                                    |           |           |           |           |          |           |
> | seed=7                             |           |           |           |           |          |           |
> | Booster (with constant $\lambda$)  | 16.31     | 92.43     | 20.46     | **87.10** | **5.58** | **16.40** |
> | Booster (with dynamic $\lambda_t$) | **13.02** | **92.78** | **13.16** | **87.10** | 13.73    | 15.60     |
> |                                    |           |           |           |           |          |           |
> | Average                            |           |           |           |           |          |           |
> | Booster (with constant $\lambda$)  | 14.31     | **92.59** | 15.88     | **86.70** | **9.06** | **16.27** |
> | Booster (with dynamic $\lambda_t$) | **8.20**  | 92.55     | **8.97**  | 85.93     | 14.93    | 15.97     |
>
>
> A key observation shared by both Booster variants is that the harmful score exhibits high variance across different random seeds, which justifies our multi-seed experimental protocol. With this breakdown, we observe that for SST2 and AGNEWS, Booster with dynamic $\lambda_t$ consistently achieves lower harmful scores while maintaining comparable fine-tuning accuracy. For GSM8K, however, the behavior is less stable: dynamic $\lambda_t$ improves harmful score in one seed (seed=5), but Booster with constant $\lambda$ achieves lower harmful scores in the other two random seeds (seed=6/7). We believe that a larger number of random seeds would yield more conclusive results.
>
> **We hope that this detailed breakdown clarifies the reviewer's questions. We again appreciate the reviewer's support for the paper.**

---

> ### Comment · Reviewer_jv5a · 2025-11-24
> **My concern is addressed**
>
> Thanks for the rebuttal. For Booster varaints and also  for other alignment stage solutions, it is unfortunately true that their defense performance have high variance. My concern is fully addressed. I will keep an eye on the later review process of this submission.

---

### Official Review · Reviewer_d57z · 2025-11-02

**Soundness:** 2
**Presentation:** 3
**Contribution:** 2
**Rating:** 2
**Confidence:** 4

**Summary:**

The paper focused on harmful fune-tuning attacks. The proposed method regularizes the gradient contribution of harmful samples. The methodology has phases that apply precaution before fine-tuning, and also during fine-tuning with a safety-preservation learning algorithm. In the first step, it makes sure that the harmful sample gradients will be in a flat loss region. In the second stage, it applies weights to the samples during the training batch to prevent learning from harmful samples.

**Strengths:**

- Clear motivation and explanation
- The paper targets a significant problem.
- The proposed method shows greater performance than the SOTA methods in some datasets.

**Weaknesses:**

- It uses the first stage to score the samples during fine-tuning. It looks like more of a single stage, where the second stage is just an extension of the first stage.

- The paper improves the alignment stage with a refusal loss, which is already used and has shown its effectiveness in the Vaccine. There is only one ablation study given for the comparison of $L_{sharp}$ and $L_{refusal}$.

- Assuming $K^t$ to be large in benign and $K^t$ would be small in harmful is vague and not supported enough; therefore, reducing equation 4 to 5 with the given proposition is unclear.

- Solution for detecting harmful samples is ad. hoc. It shows that the model is well fit to the training data, which contains the phrase “I cannot fulfill your request”

- The authors claim that the harmful loss stays constant while the benign loss decreases. However, the third plot in Figure 2 shows that, Antibody harmful loss value is high as the SFT-Harmful loss value.

- The proposed algorithm is not showing the best performance in all the datasets and for all the models, where it also uses previous methods in their strategy.

**Questions:**

- "A flat loss landscape for harmful samples makes the instilled safety behavior more difficult to remove." Can the authors define how this makes it difficult to remove? One needs more training and more samples?

- The final parameter, $\theta$, depends on the $L_{sharp}$ which depends on $L_{harm}$. To me these two losses are contradictory and their gradients could direct two opposite directions, but the methodology ends up selecting a gradient that minimizes $L_{harm}$. Overall the model is tuned towards the harmful region, but not so harmful? In other words, how do we know a gradient $\delta_t$, shown in equation 3, exists when the gradient of both losses points to exactly opposite directions?

- How is the matrix multiplication is performed in equation 6 with the given dimensionalities?

---

> ### Author Response · Authors · 2025-11-21
> **Author Rebuttal (I): Clarification about the proposed alignment stage defense**
>
> We thank the reviewer for their constructive analysis. We hope that we can clarify the reviewer's concern below.
>
> > ### **W1:** It uses the first stage to score the samples during fine-tuning. It looks like more of a single stage, where the second stage is just an extension of the first stage.
>
> Our method (1) computes the weights in the **second stage, not the first stage**, and (2) the proposed weighting mechanism in the second stage has **a clear motivation and makes a significant contribution to Antibody’s performance, rather than being just an extension of the first stage**. The following discussion clarifies these points.
>
> The FTaaS setting consists of two stages: the *alignment stage* and the *fine-tuning stage*. In the alignment stage, the service provider performs safety alignment and makes a model available for fine-tuning. In the fine-tuning stage, the service provider fine-tunes this safety-aligned model on the user-submitted dataset $\mathcal{D}\_{\text{task}}$. Since $\mathcal{D}\_{\text{task}}$ can be poisoned (benign samples mixed with harmful ones), our proposed weighting mechanism in the second stage **encourages the model to learn from benign data and makes it resistant to learning from harmful data**, resulting in a model that both preserves safety and performs well on the user-defined task. **Furthermore, the weights computed by our method change dynamically as the model is updated on $\mathcal{D}_{\text{task}}$ in the fine-tuning phase; they are not computed once from only the initial safety-aligned model.**
>
> Table 4. Ablation on our proposed components. Each row is a cumulative addition to the previous one. The values in parentheses indicate the performance change from the SFT baseline.
>
> | Method | **GSM8K HS** ↓ | **GSM8K FA** ↑ |
> | --- | --- | --- |
> | SFT | 23.94 | 10.90 |
> | + Align with $\lambda_t \, \mathcal{L}_{\text{sharp}}(\theta_t)$ | 13.02 ($-10.92$) | 16.00 ($+5.20$) |
> | **+ Fine-tune with $w_{\theta_t}$** | 4.44 ($-19.50$) | 13.83 ($+2.93$) |
> | + Align with $\mathcal{L}_{\text{refusal}}(\theta_t)$ (Antibody) | 1.24 ($-22.70$) | 15.07 ($+4.17$) |
>
> Empirically, the second-stage solution of Antibody (bolded in the table below) provides a **significant gain in both defense and learning performance**, as shown in Table 4 in our paper (shown above).
>
> > ###  **W2:** The paper improves the alignment stage with a refusal loss, which is already used and has shown its effectiveness in the Vaccine. There is only one ablation study given for the comparison of $L\_{sharp}$ and $_{refusal}$.
>
> We thank the reviewer for pointing out the connection of our method with Vaccine [1]. We are happy to clear the confusion regarding our refusal loss term and the loss used in Vaccine.
>
> Our refusal loss term $\mathcal{L}\_{\text{refusal}}$ is **different** from the loss used in Vaccine. Vaccine is an alignment stage defense solution that aims to mitigate the harmful embedding drift that happens during harmful fine-tuning. In the alignment stage, Vaccine proposes to compute the loss over the safety alignment dataset after **adding a perturbation to the inner hidden embeddings of the model parameters $\theta$**. While our refusal loss computes the loss over the **harmful perturbation model $\theta_{\text{pert}}$**, by optimizing this objective, we want to simulate a situation in the fine-tuning stage where the model being updated using harmful samples can still maximize the refusal response for the harmful samples. Consequently, the weight $w_{\theta_t}$ for a harmful sample will be low, as the weight is inversely proportional to the likelihood of generating a refusal response.
>
> ---
> **References:**
>
> [1] Vaccine: Perturbation-aware Alignment for Large Language Models against Harmful Fine-tuning Attack. (NeurIPS 2024).

---

> ### Author Response · Authors · 2025-11-21
> **Author Rebuttal (II): Discussion about reducing Equation (4) to Equation (5)**
>
> > ### **W3:** Assuming $K^t$ to be large in benign and $K^t$ would be small in harmful is vague and not supported enough; therefore, reducing equation 4 to 5 with the given proposition is unclear.
>
> We thank the reviewer for this question. To better answer this question, we split W3 into two parts and address them separately.
>
>
> > Assuming $K^t$ to be large in benign and $K^t$ would be small in harmful is vague and not supported enough ...
>
>
> The term $\mathcal{K}^t$ measures the similarity between the token-level features of two samples: the update sample at step $t$ and the test sample whose likelihood we track. Our argument is that, **for a benign test sample, $\vert| \mathcal{K}^t \vert|_F$ is small when the update sample is harmful and larger when the update sample is also benign.** Intuitively, harmful and benign samples correspond to different tasks, so harmful–benign pairs have weak gradient alignment (small $\vert| \mathcal{K}^t \vert|_F$), whereas benign–benign pairs are more aligned (large $\vert| \mathcal{K}^t \vert|_F$) since they are belong to same task. This argument is consistent with prior analyses of LLM finetuning [2].
>
>
> > ... therefore, reducing equation 4 to 5 with the given proposition is unclear.
>
> We want to clarify that **the step from Equation (4) to Equation (5) *does not rely* on the eNTK term $\mathcal{K}^t$, but rather on the fact that the gradients of harmful samples are small at the beginning of the fine-tuning stage due to our flatness regularization in the alignment stage**, leading to the gradient of the loss over harmful examples (i.e., $\sum_{\left(x_{i},y_{i}\right)\,is\,harmful}\nabla\ell_{\theta_{t}}\left(x_{i},y_{i}\right)$) negligible. The subsequent proposition in the paper is used only as a tool to analyze the learning dynamics of the mini-batch update. Additionally, using Prop. 4.2, we want to demonstrate that with the update rule in Equation (5), the $\Delta \ell(x_o, y_o)$ is negligible for harmful examples $(x_o, y_o)$. As a result, the LLM model **does not change** its effect/behavior on the harmful examples, while **actively learning** from benign examples. Finally, as pointed out by Prop. 4.3, our weighting mechanism further strengthens this property by setting small weights on potentially harmful examples in the fine-tuning stage for **maximally reducing** the effect of the harmful examples on the update of the LLM model.
>
> We have **added Figure 4 to Appendix F.6 that demonstrates the effect of our flatness regularization in the alignment stage and validates the claim that harmful gradients are small after aligning with Antibody.** Figure 4 in the appendix shows that, for the model aligned with Antibody, the gradient norms on harmful samples are clearly smaller than those of the model aligned with standard SFT, as indicated by the left-shifted distribution. Moreover, under Antibody, harmful gradient norms are also smaller than benign gradient norms, indicating that harmful examples contribute much less to the overall update direction than benign ones. By contrast, under standard SFT alignment, harmful gradients remain large relative to benign gradients, so harmful samples still exert a strong influence on the update direction.
>
> ---
> **References:**
>
> [2] Learning Dynamics of LLM Finetuning (ICLR 2025, Outstanding Paper Award).

---

> ### Author Response · Authors · 2025-11-21
> **Author Rebuttal (III): Clarification about the proposed fine-tuning stage defense**
>
> > ### **W4:** Solution for detecting harmful samples is ad. hoc. It shows that the model is well fit to the training data, which contains the phrase “I cannot fulfill your request”
>
> We thank the reviewer for raising this concern. We would like to clarify that our approach is **not an ad hoc detection heuristic**.
>
> First, our method does **not perform explicit harmful-sample detection nor apply any cut-off method to remove samples**. Instead, we introduce a weighting mechanism (Eq. (7)) that is applied to all samples in the fine-tuning dataset. The weighting rule is motivated by a clear objective: to encourage learning from benign data while suppressing updates driven by harmful data. It leverages the model's refusal behavior learned in the alignment stage as a continuous signal, rather than relying on any specific refusal phrase.
>
> As shown in Figure 2 (left and middle), **unseen harmful prompts in the fine-tuning phase consistently receive low weights**, whereas benign prompts receive much higher weights, indicating that the mechanism generalizes beyond the training phrases and is not simply "well fit to the training data.". In Appendix F.9, we further conduct the out-of-distribution fine-tuning attack with harmful poison data from PureBad [3], which is from a different distribution than BeaverTails [4]. The separate score distributions of the harmful and benign data before and after the fine-tuning attack in Figure 5 (Appendix F.9) show that **our proposed weighting mechanism is able to generalize to other harmful datasets.**
>
> Finally, our Experiments section and Figure 1 demonstrate that this weighting mechanism (together with our other proposed defenses) yields **significantly lower harmful scores** on unseen harmful samples, providing further evidence that the method is not merely fitting to the training data.
>
> > ### **W5:** The authors claim that the harmful loss stays constant while the benign loss decreases. However, the third plot in Figure 2 shows that, Antibody harmful loss value is high as the SFT-Harmful loss value.
>
> Figure 2 (right) **does show that the harmful loss of Antibody is stable while the benign loss decreases**, consistent with our claim. The key point in this figure is the training dynamics, not the absolute value of the harmful loss. For Antibody, the harmful loss remains approximately constant throughout the fine-tuning stage, while the benign loss decreases. This indicates that the model is not fitting the harmful data during fine-tuning, in contrast to the SFT baseline, whose harmful loss continues to decrease.
>
> This behavior is exactly what our flatness regularization in the alignment stage is designed to achieve. By enforcing sharpness regularization on harmful samples, we drive the model into a **flat region of the harmful-loss landscape**, where **the gradient of the harmful loss is small** (refer to Figure 4 to Appendix F.6). As a result, during the fine-tuning phase, updates on harmful samples have little effect (small gradients), while benign samples still provide informative gradients that reduce the benign loss.
>
> Regarding the reviewer's comment about the absolute value of Antibody's harmful loss, we emphasize that at the end of the fine-tuning process, the harmful loss of Antibody is **lower than that of SFT**, as shown in Figure 2 (right).
>
> ---
> **References:**
>
> [3] Fine-tuning Aligned Language Models Compromises Safety, Even When Users Do Not Intend To! (ICLR 2024)
>
> [4] BeaverTails: Towards Improved Safety Alignment of LLM via a Human-Preference Dataset (NeurIPS 2023)

---

> ### Author Response · Authors · 2025-11-21
> **Author Rebuttal (IV): Concern about performance**
>
> > ### **W6:** The proposed algorithm is not showing the best performance in all the datasets and for all the models, where it also uses previous methods in their strategy.
>
> We thank the reviewer for this observation and agree that Antibody does not achieve the best performance on every model and every dataset. We first clarify that Antibody does **not use Vaccine’s loss function** in its algorithm, as detailed in our response to Weakness 2.
>
> Although Antibody is not uniformly best across all settings, it provides strong and consistent gains on the key safety metric while preserving benign performance. On the main benchmark (Table 1), Antibody achieves the best harmful score on three out of four datasets and the best fine-tuning accuracy on two out of four datasets. Across different model architectures (Table 2), Antibody attains the best harmful score on all three models and the second-best fine-tuning accuracy. Moreover, Figure 1 shows that existing methods exhibit a clear trade-off between safety and fine-tuning accuracy, whereas **Antibody achieves a superior safety–utility trade-off by attaining stronger safety with comparable or better fine-tuning accuracy.**
>
>
> Finally, we note that it is inherently challenging for any single defense to dominate across all models and datasets. **Prior state-of-the-art methods, such as Booster [5] and Lisa [6], also report that they do not outperform all baselines in every setting.** In this context, Antibody’s consistent improvements in harmful scores, together with its competitive fine-tuning accuracy and favorable safety–utility trade-off, demonstrate its practical effectiveness.
>
> ---
> **References:**
>
> [5] Booster: Tackling Harmful Fine-tuning for Large Language Models via Attenuating Harmful Perturbation. (ICLR 2025).
>
> [6] Lisa: Lazy Safety Alignment for Large Language Models against Harmful Fine-tuning Attack. (NeurIPS 2024)

---

> ### Author Response · Authors · 2025-11-21
> **Author Rebuttal (V): Clarification about the flatness regularization defense**
>
> > ### **Q1:** "A flat loss landscape for harmful samples makes the instilled safety behavior more difficult to remove." Can the authors define how this makes it difficult to remove? One needs more training and more samples?
>
> In our framework, this means that, **under a fixed fine-tuning budget (number of steps, learning rate, and harmful samples), harmful fine-tuning updates have a smaller effect on the harmful loss or the harmful score than standard SFT**. Equivalently, an attacker would need either many more harmful updates or much larger parameter changes to degrade safety compared to the non-flat case. Intuitively, once the model has been moved into a flat region of the harmful loss during the alignment stage, the harmful gradients in the fine-tuning stage are small, so each update only barely changes how the model behaves on harmful prompts.
>
> We have validated the robustness of our method under several stronger attack settings: more harmful samples (Figure 1) or higher poison ratios (Table 3), longer training (more epochs; Figure 3, left), and more aggressive updates on harmful data via higher learning rates (Figure 3, right).

---

> ### Author Response · Authors · 2025-11-21
> **Author Rebuttal (VI): Clarification about the flatness regularization formulation**
>
> ### **Q2:** Clarification about flatness regularization formulation
>
> We thank the reviewer for this insightful question. For clarity, we split Q2 into two parts and address them separately.
>
> > "... but the methodology ends up selecting a gradient that minimizes $L_{harm}$. Overall the model is tuned towards the harmful region, but not so harmful?":
>
> As shown in Equation (3) and in Proposition 4.1, the update direction $\delta\_t$ is chosen to decrease both $\mathcal{L}\_{\text{align}}$ and $\mathcal{L}\_{\text{sharp}}$, **not to minimize $\mathcal{L}\_{\text{harm}}$**. Even though the definition of $\mathcal{L}\_{\text{sharp}}$ involves $\mathcal{L}\_{\text{harm}}$, specifically:
>
> $\mathcal{L}\_{\text{sharp}}(\theta') \triangleq \mathcal{L}\_{\text{harm}}(\theta') - \min\_{\phi \in \mathcal{B}\_\rho(\theta')} \mathcal{L}\_{\text{harm}}(\phi),$
>
> it **also includes the subtraction term $- \min\_{\phi\in\mathcal{B}\_{\rho}(\theta')} \mathcal{L}\_{\text{harm}}(\phi)$**. This turns $\mathcal{L}\_{\text{sharp}}$ into a *flatness* measure: it is small when $\mathcal{L}\_{\text{harm}}$ cannot be further reduced within the $\rho$-ball $\mathcal{B}\_\rho(\theta')$, i.e., when the harmful loss is locally flat around $\theta'$.
>
> Consequently, minimizing $\mathcal{L}\_{\text{sharp}}$ does **not drive the model towards regions of low harmful loss**, but towards regions where the harmful loss is locally insensitive to parameter changes. At the same time, minimizing $\mathcal{L}\_{\text{align}}$ explicitly encourages high loss on harmful completions (in other words: low loss on refusal completions). Thus, the method is not "tuning the model towards the harmful region, but not so harmful". Instead, it finds parameters that are (i) strongly refusal-aligned and (ii) located in a flat region of the harmful loss landscape, making the safety behavior robust to subsequent harmful fine-tuning.
>
>
> >"... how do we know a gradient $\delta\_t$ shown in equation 3, exists when the gradient of both losses points to exactly opposite directions?"
>
> Regarding the case where the two gradients in Equation (3) appear to be in opposite directions: let $g\_t = \nabla\_{\theta} \mathcal{L}\_{\text{align}}(\theta\_t)$ and $h\_t = \nabla\_{\theta} \mathcal{L}\_{\text{sharp}}(\theta\_t)$. When $g\_t^{\top} h\_t$ is negative for a given batch (i.e., the two gradients strongly disagree), the formula in Theorem 4.1 implies that the numerator $a\_t - g\_t^{\top} h\_t$ increases and thus $\lambda\_t$ becomes larger. In this situation, $\delta\_t^{*} = g\_t + \lambda\_t h\_t$ is a weighted sum of two almost opposite vectors, so the two terms partially cancel each other. As a result, the overall update $\delta\_t$ becomes smaller and more conservative: the method avoids making a large step in the case of highly conflicting gradients.
>
> In practice Figure 2 (right) illustrate that the update rule in Theorem 4.1 sucessfully optimize both $\mathcal{L}\_{\text{align}}$ and $\mathcal{L}\_{\text{harmful}}$ objectives: at the beginning of the fine-tuning phase, the harmful loss on harmful completions remains high (the cross-entropy loss is far from 0 and higher than 1), indicating that the sequence of $\delta\_t$ updates has indeed optimized the alignment objective $\mathcal{L}\_{\text{align}}$ in the alignment stage. Moreover, the harmful loss stays approximately constant throughout the fine-tuning phase, which is consistent with successfully optimizing $\mathcal{L}\_{\text{sharp}}$ so that harmful loss becomes locally flat and resistant to further decrease under harmful fine-tuning (while we acknowledge that the weighted loss used in the fine-tuning phase also contributes to this stability).

---

> ### Author Response · Authors · 2025-11-21
> **Author Rebuttal (VII): Explain matrix multiplication in Equatiion (6)**
>
> > ### **Q3:** How is the matrix multiplication is performed in equation 6 with the given dimensionalities?
>
> The matrix multiplication in Equation (6) is well-defined given the stated dimensionalities. Concretely, the indexing operations extract lower-dimensional slices as follows:
> - $[\mathcal{A}^t]_m$: Although $\mathcal{A}^t \in \mathbb{R}^{1 \times V \times M}$ is a 3D tensor. Fixing the index $m$ along the third dimension yields a row vector of shape $1 \times V$.
> - $[\mathcal{K}^t]_{m,l}$: The tensor $\mathcal{K}^t \in \mathbb{R}^{V \times V \times M \times L}$ is a 4D tensor. Fixing the output index $m$ along the third and index $l$ along the fourth dimensions yields a matrix of shape $V \times V$.
> - $[\mathcal{G}^t]_l$: The matrix $\mathcal{G}^t \in \mathbb{R}^{V \times L}$ becomes a column vector of shape $V \times 1$ when we fix the index $l$ along the second dimension.
>
> The product in Equation (6) is then a standard sequence of matrix multiplications over the vocabulary dimension: $(1 \times V) \times (V \times V) \times (V \times 1) = (1 \times 1)$, yielding a scalar for each term in the summation.

---

> ### Comment · Reviewer_d57z · 2025-11-26
> **Response to Rebuttal**
>
> Thank you for the detailed rebuttal; some of my concerns are addressed. Therefore, I will increase my score. I will keep reviewing the discussion and continue asking about the unclear points.

---

> ### Author Response · Authors · 2025-11-26
> **Thank the reviewer for the response and score update**
>
> Thank you for your feedback and for raising the score. We appreciate your continued engagement with our work. Regarding the remaining unclear points, **could you please specify which concerns you feel were not fully addressed in our rebuttal?** We are eager to provide further clarification to resolve any outstanding issues during the discussion period.

---

### Author Response · Authors · 2025-12-01
**Author summary of the discussion period**

Dear Area Chair and Reviewers,

We sincerely thank the reviewers for their constructive comments and active engagement during the discussion period. The initial reviews were encouraging, starting with ratings of 8 (accept, good paper), 2 (reject, not good enough), and 6 (weak accept). **Following the rebuttal phase, during which we addressed the raised issues, two reviewers (d57z and xQyK) agreed to increase their ratings.** Notably, Reviewer jv5a (confidence 5) noted that this is a very solid work and has championed the paper during the rebuttal process.

Below are the summaries of our interactions with each reviewer.

**Reviewer d57z:** The reviewer acknowledged that our method has clear motivation and achieves SOTA performance in many settings, though they initially expressed concern regarding the formulation and performance against baselines.
- During the rebuttal: We provided detailed written clarifications regarding the formulation of the proposed alignment and defense stages. We backed our arguments with experimental visualizations of the harmful gradient norm (Appendix F.6) and a study on the robustness of weight allocation under out-of-distribution (OOD) attacks (Appendix F.9). **The reviewer agreed that several of their concerns were addressed and subsequently agreed to increase their score.** Reviewer d57z remained open to further discussion, but the comment period unfortunately ended before we could conclude.

**Reviewer jv5a:** The reviewer commented that our method is fundamental and elegant, showing superior performance compared to baselines. Their main concern focused on the writing and the relationship of our work to existing literature (specifically Booster).
- During the rebuttal: We updated the paper to explicitly discuss the relationship between the proposed method and Booster in Section 4.1 and other related work in Appendices A, B, and C. We addressed the reviewer's questions via ablation studies on the step-adaptive regularizer intensity $\lambda_t$ (Appendix F.7) and refusal loss $\mathcal{L}_{\text{refusal}}$ (Appendix F.8), alongside detailed results on random seeds and OOD robustness (Appendix F.9). **The reviewer agreed that we fully addressed their concerns and decided to maintain their positive score of 8 (accept, good paper).**

**Reviewer xQyK:** The reviewer found our core idea interesting and results promising, but requested clarification on method details, overhead, and performance with respect to the size of the safety alignment dataset.
- During the rebuttal: We updated Section 4.1 to provide a better interpretation of Theorem 4.1, highlighted the system analysis present in the original submission (Appendix F.4), and added an analysis regarding the safety alignment dataset size (Appendix F.5). **The reviewer agreed that their concerns were mostly addressed and increased their score.**

We believe that the rebuttal process has significantly strengthened our paper and that most concerns have been successfully addressed. We again thank the reviewers for their insightful reviews and acknowledgment of our work.

---

### Meta-Review · Area_Chair_VAWQ · 2026-01-01

**Summary:**

This paper proposed Antibody: Strengthening Defense Against Harmful Fine-Tuning for Large Language Models via Attenuating Harmful Gradient Influence.

This paper received mixed review scores (2,6,8). The authors provided detailed responses, and reviewers actively engaged in discussions. After the rebuttal, both reviewer d57z (score 2) and reviewer xQyK (score 6) agreed to raise their scores.

The concerns reviewers had raised included clarification of the proposed method, extensive discussion with related work, and more ablation studies. The authors' responses resolved a lot of clarification questions, particularly for reviewer d57z that might flip the original assessment. Overall, there does not seem to be a major flaw in the paper, and it provides an interesting study on the important problem of fine-tuning attack and defense. I believe the reviewers’ feedback would help improve the draft, and I would particularly encourage authors to keep iterating on the paper presentation for clarity.

**Reviewer Concerns:**

The concerns reviewers had raised included clarification of the proposed method, extensive discussion with related work, and more ablation studies. The authors' responses resolved a lot of clarification questions, particularly for reviewer d57z that might flip the original assessment. Overall, there does not seem to be a major flaw in the paper, and it provides an interesting study on the important problem of fine-tuning attack and defense.

**Reviewer Scores:**

This paper received mixed review scores (2,6,8). The authors provided detailed responses, and reviewers actively engaged in discussions. After the rebuttal, both reviewer d57z (score 2) and reviewer xQyK (score 6) agreed to raise their scores.

---

### Decision · Program_Chairs · 2026-01-26

Accept (Poster)